# Thought Anchors: Which LLM Reasoning Steps Matter?

## Abstract

Current frontier large-language models rely on reasoning to achieve state-of-the-art performance. Many existing interpretability methods are limited in this area, as standard methods have been designed to study single forward passes of a model rather than the multi-token computational steps that unfold during reasoning. We argue that analyzing reasoning traces at the sentence level is a promising approach to understanding reasoning processes. We introduce a black-box method that measures each sentence's counterfactual importance by repeatedly sampling replacement sentences from the model, filtering for semantically different ones, and continuing the chain of thought from that point onwards to quantify the sentence's impact on the distribution of final answers. We discover that certain sentences can have an outsized impact on the trajectory of the reasoning trace and final answer. We term these sentences *thought anchors*. These are generally planning or uncertainty management sentences, and specialized attention heads consistently attend from subsequent sentences to thought anchors. We further show that examining sentence-sentence causal links within a reasoning trace gives insight into a model's behavior. Such information can be used to predict a problem's difficulty and the extent different question domains involve sequential or diffuse reasoning. As a proof-of-concept, we demonstrate that our techniques together provide a practical toolkit for analyzing reasoning models by conducting a detailed case study of how the model solves a difficult math problem, finding that our techniques yield a consistent picture of the reasoning trace's structure. We provide an open-source tool (anonymous-interface.com) for visualizing the outputs of our methods on further problems. The convergence across our methods shows the potential of sentence-level analysis for a deeper understanding of reasoning models.

## 1 Introduction

Training large language models to reason with chain-of-thought (Reynolds & McDonell, 2021; Nye et al., 2021; Wei et al., 2023) has significantly advanced their capabilities (OpenAI, 2024). The resulting reasoning traces are regularly used in safety research (Baker et al., 2025; Shah et al., 2025), but there has been little work adapting interpretability methods to this new paradigm ((Venhoff et al., 2025; Goodfire, 2025). Traditional *mechanistic interpretability* (Olah et al., 2020; Olah, 2022) methods often focus on a single forward pass of the model, understanding how layer-by-layer activations (Wang et al., 2022; Heimersheim & Janiak, 2023). However, this too fine-grained for autoregressive reasoning models, which consume their own output tokens.

Interpretability generally research aims to find the causes of a model's behavior, to decompose a model into smaller parts, and to map the mechanisms linking intermediate states to a model's final output. During reasoning, these intermediate states correspond to text in the chain-of-thought (CoT). Our goal in this paper is to shed light about the computations being performed by CoT text and to identify high-level principles regarding the structure of CoT. In addition, we present methods for measuring how particular pieces of text drive the model's final answer and influence one another.

For reasoning models, we propose that chain-of-thought traces can be decomposed into multi-token reasoning steps. We operationalize reasoning steps in terms of *sentences*. Compared to individual tokens, sentences are more coherent and often coincide with reasoning steps extracted by an LLM (Venhoff et al., 2025; Arcuschin et al., 2025), and recent work suggests that sentence-

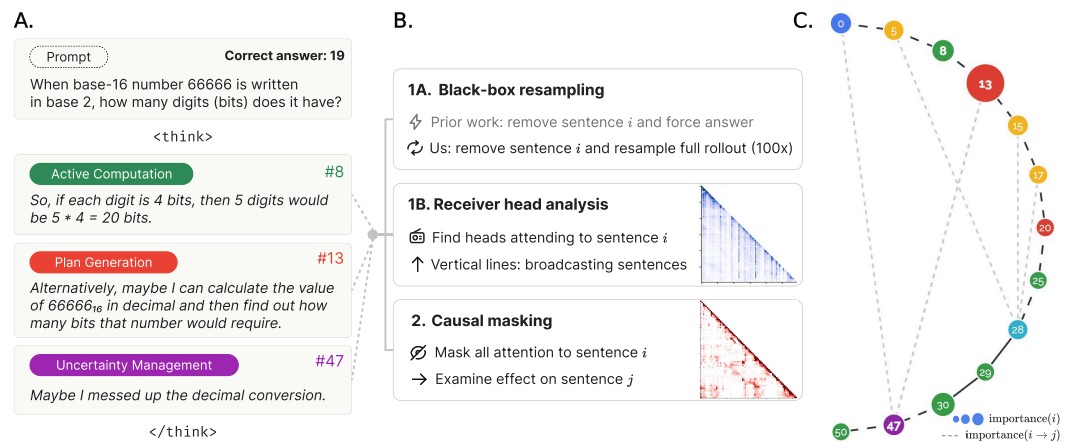

Figure 1: Summary of our methods for principled attribution to important sentences in reasoning traces. **A.** An example reasoning trace with sentences labeled per our taxonomy. **B.** Our proposed methods are: black-box resampling, receiver heads, and attention suppression. **C.** A directed acyclic graph among sentences prepared by one of our techniques, made available open source.

ending punctuation frequently acts as an information-compressing boundary (Razzhigaev et al., 2025; Chauhan et al., 2025). While a reasoning steps sometimes may be better seen as a sub-sentence phrases or multiple sentence paragraph, we treat sentence segmentation as a robust starting point, and we provide evidence validating the efficacy of this approach for studying CoTs.

We argue that CoTs are characterized by *thought anchors*: critical points in the CoT that guide the reasoning trace's trajectory. We provide evidence for this type of anchoring based on black-box evidence from resampling and white-box evidence based on attention patterns. By measuring the causal dependencies between sentences via a masking approach, we further show how a CoTs wider computational structure can be interpreted. These measures go beyond just reading a CoT's text, providing a principled foundation for interpretability that sidesteps disputes about the "faithfulness" of CoT text (Turpin et al., 2023; Korbak et al., 2025).

Section 2 and Section 3 provide evidence for the existence of particularly impactful sentences and introduce a black-box method for measuring the counterfactual impact of a sentence on the model's final answer. Our method repeatedly resample reasoning traces from the start of each sentence. Based on resampling data, we can quantify the counterfactual impact of each sentence on the likelihood of any final answer. We find that planning sentences systematically initiate computations leading to some answer and play a role distinct from sentences performing computations necessary for the answer but which are predetermined. Section 4 adds a white-box method for evaluating importance based on the sentences most attended. Our analyses reveal "receiver" heads that narrow attention toward particular past "broadcasting" sentences. This provides a mechanistic measure of importance, whose findings converge with our resampling technique.

Section 5 and Section 6 present a method mapping the wider structure of a CoT in terms of the causal dependencies between pairs of sentences. For each sentence in a trace, we mask all attention to it from subsequent tokens or by removing the sentence entirely. We then measure the effect on subsequent token logits (KL divergence) compared without masking. Averaging token effects by sentence, this strategy measures each sentence's direct causal effect on each subsequent sentence.

Applying these techniques, our work suggests that analyzing reasoning through sentence-level units introduces new domains through which reasoning models can be understood. Our work also opens the door to more precise debugging of reasoning failures, identification of sources of unreliability, and the development of techniques to enhance the reliability of reasoning models.

## 2 CASE STUDY ON SENTENCE IMPORTANCE

We demonstrate in this section how sentences can be important by influencing the downstream reasoning trace: a sentence is important if changing it would alter the subsequent CoT and the final answer. We study this by conditioning the model on the CoT up to a given sentence, repeatedly sampling the model continuing from that point, and comparing the resulting answer distributions across different continuation points. We compare our strategy to the existing standard approach for evaluating the impact of parts of a CoT by interrupting the model and forcing it to output its final answer from that point. Here we a detailed case study to show how this standard approach ignores sentences that are causally important by influencing the downstream CoT, but that our technique captures this. In addition, we provide evidence on the utility of specifically examining sentences.

### 2.1 MODEL AND DATASET

Our analyses of sentence importance in this section along with Sections section 3, section 4, and section 5 employ the DeepSeek R1-Distill Qwen-14B model, using a temperature of $0.6$ and a top-p value of $0.95$(DeepSeek, 2025). For the present case study we focus on just one problem from the MATH dataset (Hendrycks et al., 2021).

### 2.2 FORCED ANSWER IMPORTANCE

Earlier work has measured sentence importance by forcing a model to answer before completing its reasoning trace (Lanham et al., 2023a; Radhakrishnan et al., 2023; Wang et al., 2025; Tanneru et al., 2024; Parcalabescu & Frank, 2024). We compared our approach to this existing technique: For each sentence in a CoT, we interrupt the model and append text, inducing a final output ("`Therefore, the final answer is \boxed{`"). This was done 100 times at each sentence position.

### 2.3 IMPORTANCE VIA RESAMPLING

A limitation of the forced-answer approach is that a sentence $S$ may be necessary for some final answer but is consistently produced by the LLM late in the reasoning trace (e.g., a reliable arithmetic statement). Thus, forced answer accuracy will be low for all sentences before $S$, precluding earlier step importance from being assessed.

Our approach evaluates importance by examining how a sentence may guide downstream sentences. Consider a rollout consisting of sentences $S_1, S_2, \ldots, S_i, \ldots, S_M$ and a final answer $A$. We can use resampling to capture the extent sentence $S_i$ influences $A$. Specifically, for a given sentence $S_i$, we generate a distribution over final answers by generating 100 rollouts both without sentence $S_i$ (rollouts of the form $S_1, S_2, \ldots, S_{i-1}, T_i, \ldots, T_N, A'_{S_i}$), and another distribution with sentence $S_i$ (rollouts of the form $S_1, S_2, \ldots, S_i, U_{i+1}, \ldots, U_M, A_{S_i}$).

### 2.4 CASE STUDY

We first investigate the efficacy of our sentence importance technique by applying it to one problem: *"When the base-16 number $66666_{16}$ is written in base 2, how many base-2 digits (bits) does it have?"* (MATH Problem 4682; see Section A.1 for the CoT transcript). The resampling data shows that from sentences 6-12, expected accuracy steadily declines, but sentence 13 causes accuracy to drastically increase (indicated by the navy and red circles in Figure 2A).

The large accuracy fluctuation motivates inspection of this part of the CoT. The model initially considers that $66666_{16}$ contains five base-16 digits, and any base-16 digit can be represented with four base-2 digits. Thus, the model considers the answer: 20 bits. However, this overlooks that $6_{16}$ is $110_2$ rather than $0110_2$ due to the leading zero. Interestingly, Sentence 12 mentions *"checking if there's any leading zero that might affect the bit count,"* yet Sentence 12 lowers the expected accuracy. The uplift comes from Sentence 13, where the model decides to *"calculate the value of $66666_{16}$ in decimal"* (see resample alternatives in Section A.2). Downstream reasoning computes the decimal value of $66666_{16}$ and converts it to binary to arrive at the correct answer: 19 bits. The key role of Sentence 13 is missed if examining forced-accuracy importance (Figure 2A). This case study provides initial evidence on how resampling identifies moments in a CoT where impactful plans are set.

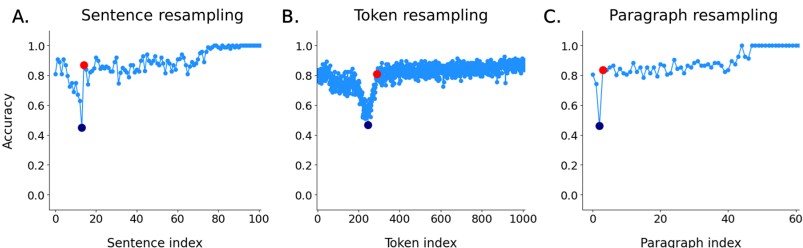

Figure 2: Accuracy over 100 rollouts at each (**left**) token, (**middle**) sentence, or (**right**) paragraph. Navy and red circles border the most importance sentence (Sentence 13) and are plotted in each graph as a reference. For the token graph, resampling was only done on the first 1,000 tokens of the CoT.

Further tests show the efficacy of specifically examining sentences. The sentence-level resampling data mirrors the patterns seen resampling tokens but at a fraction of the cost (Figure 2B), whereas resampling paragraphs leads to meaningfully less resolution (Figure 2C).

## 3 CONSISTENT PATTERNS IN SENTENCE IMPORTANCE

We now move to investigating whether particularly important sentences are a consistent theme across CoTs, and whether they can be systematically related to sentence content. We formalize our resampling approach into an importance score that can be compared across sentences and aggregated across problems, and we apply analyzing CoTs for challenging MATH questions. This lets us analyze how importance varies across different sentence types (e.g., planning statements) and to contrast what our proposed measure captures compared to the typical forced-importance measure. In doing so, we move beyond a single case study to characterize patterns of sentence importance in model reasoning.

### 3.1 DATASET

As for the case study, we examine problems from the MATH dataset. As our analysis hinges on variability in final responses, so we target 20 challenging but doable questions that are correctly solved 25-75% of the time, identified by testing on 1,000 problems 10 times each. For each selected problem, we generated one correct and one incorrect reasoning trace, producing 40 responses. The average response is 144.2 sentences (95% CI: [116.7, 171.8]) and 4208 tokens (95% CI: [3479, 4937]). We focus only on sentences before the model has converged on an answer (i.e., after which it gives the same response in >98% of resamples). In Section B, we provide results from applying our techniques to the R1-Distill-Llama-8B model.

### 3.2 SENTENCE TAXONOMY

To more systematically test whether reasoning is characterized by key sentences with outsized impacts, we organized sentences into different categories and measured their causal impacts. We adopted the framework by Venhoff et al. (2025), which defines distinct reasoning functions within a reasoning trace. We specify eight categories (see examples and frequencies in Section C):

1. **Problem Setup:** Parsing or rephrasing the problem
2. **Plan Generation:** Stating or deciding on a plan of action, meta-reasoning
3. **Fact Retrieval:** Recalling facts, formulas, problem details without computation
4. **Active Computation:** Algebra, calculations, or other manipulations toward the answer
5. **Uncertainty Management:** Expressing confusion, re-evaluating, including backtracking
6. **Result Consolidation:** Aggregating intermediate results, summarizing, or preparing
7. **Self Checking:** Verifying previous steps, checking calculations, and re-confirmations
8. **Final Answer Emission:** Explicitly stating the final answer

Each sentence in the analyzed response is assigned to one of these categories using an LLM-based auto-labeling approach (detailed in Section D). Categories that rarely appear are omitted from the figures below. Residual-stream probes accurately distinguish categories (see Section E).

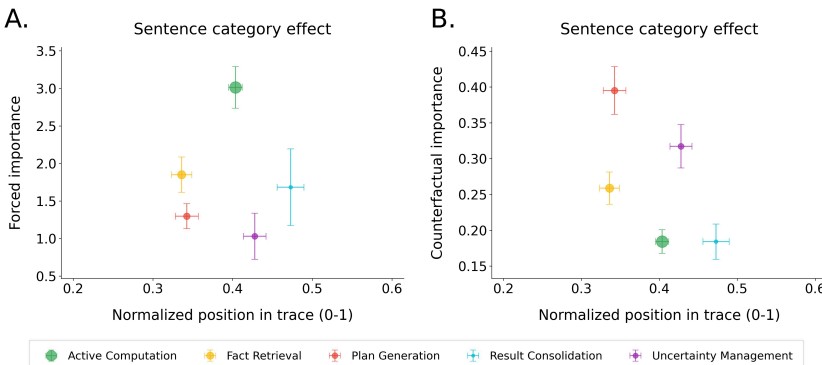

Figure 3: Plots show each sentence category for (**A**) forced-answer importance and (**B**) counterfactual importance; 5 most common sentence types shown (see Section J). The x-axis shows the sentence's average position in a reasoning trace to show this does not explain the difference in importance.

### 3.3 COUNTERFACTUAL IMPORTANCE

We additionally formalize our approach to quantifying importance in a manner that can be applied to any problem, including ones with any number of possible outcomes. We present two measures:

1. **Resampling importance.** We can compute the KL Divergence between the final answer distributions when conditioning up to $S_i$, $p(A_{S_i})$, or up to the prior sentence, $p(A'_{S_i})$, i.e., importance$_r := D_{KL}[p(A'_{S_i})||p(A_{S_i})]$, providing a measure of how much sentence $S_i$ changes the answer. We call this *resampling importance*. Because we resample all steps after a given sentence $S_i$, we avoid the aforementioned limitation of forced-answering.
2. **Counterfactual importance.** The problem with resampling importance is that if $T_i$ is identical or similar to $S_i$ then we do not get much information about whether $S_i$ is important or not. Therefore, we write $S \not\approx T$ if two sentences $S$ and $T$ are dissimilar, defined as having embeddings with a cosine similarity less than the median value across all sentence pairs in our dataset; see Section F for details and evidence the below findings remain consistent across other thresholds. We can define *counterfactual importance* by conditioning on $T_i \not\approx S_i$; i.e., importance $:= D_{KL}[p(A'_{S_i}|T_i \not\approx S_i)||p(A_{S_i})]$. Our KL divergence analyses include $\epsilon = 10^{-9}$ to avoid division by zero, but our findings remain consistent if performed using additive smoothing ($\alpha = 0.5$ or $1.0$; Section G).

Our analyses below continue with **counterfactual importance**, comparing it to *forced answer importance* also computed based on final-answer KL divergence. Relative to resampling importance, counterfactual importance conditions on semantically different resamples, reducing overdetermination and better isolating a sentence's causal influence. Resampling importance remains a complementary metric that captures how far a prediction deviates from the model's average behavior and may be preferable when that notion of deviation is primary; we compare the two in Section H.

### 3.4 RESULTS

*Plan generation* and *uncertainty management* (e.g., backtracking) sentences consistently show higher counterfactual importance than other categories like *fact retrieval* or *active computation* (see Figure 3B). This supports the view that high-level organizational sentences anchor, organize, and steer the reasoning trajectory. These findings deviate from the analysis of forced answer importance, which instead implicates *active computation* as producing the greatest distributional shifts (Figure 3A). The forced-answer approach entirely neglects the importance of planning that influences other sentences, which we argue is more meaningful for understanding the trajectory of a reasoning trace. In Section I we also provide results from an LLM judge tasked to score each sentence's importance based on reading the text. The judge likewise focuses on active computation steps and misses how a sentence influences the downstream reasoning, suggesting this is difficult to predict.

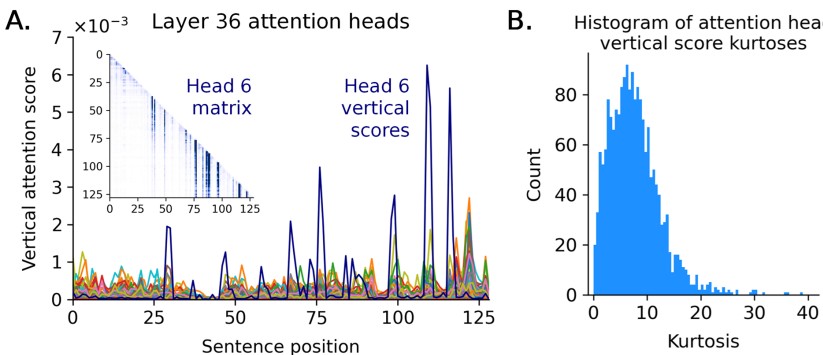

Figure 4: **A**. Lines show the vertical attention scores for each sentence by the 40 different heads in layer 36. Head 6 has been highlighted as a receiver head, and its corresponding attention weight matrix is shown for reference. Its prominent spikes cause the distribution to have a high kurtosis. **B**. Histogram of these kurtosis values across all attention heads, median across all reasoning traces.

## 4 MECHANISTIC EVIDENCE FOR SENTENCE IMPORTANCE

We hypothesize that important sentences may receive heightened attention. Although attention weights do not necessarily imply causal links, attention is a plausible mechanism by which important sentences influence subsequent sentences. We conjecture further that a focus on important sentences may be driven by specific attention heads, and by tracking such heads, we may pinpoint key sentences.

We assessed the degree that different attention heads narrow attention toward particular sentences. For each of the 40 reasoning traces produced for the MATH problems, we averaged each attention head's token-token attention weight matrix to form a sentence-sentence matrix, where each element is the mean across all pairs of tokens between two sentences. For each matrix, we computed the mean of its columns below the diagonal to measure the extent each sentence receives attention from all downstream sentences; averaged only among pairs at least four sentences apart. This generates a distribution for each head (e.g., Figure 4A), and the extent each head narrows attention toward specific sentences in general can be quantified as its distribution's kurtosis. Plotting each head's kurtosis reveals that some attention heads strongly narrow attention toward specific sentences (Figure 4B).

### 4.1 THE IDENTIFICATION OF RECEIVER HEADS

We refer to attention heads that narrow attention toward specific sentences as *"receiver heads"*. These heads are more common in later layers (Section K). To formally assess the existence of receiver heads, we tested whether some attention heads consistently operate in this role by measuring the split-half reliability of heads' kurtosis scores. We found a strong head-by-head correlation ($r = .84$) between kurtosis scores computed for half of the problems with kurtosis scores for the other half of problems. Thus, some attention heads consistently operate as receiver heads, albeit with some heterogeneity across responses in which heads narrow attention most.

Receiver heads usually direct attention toward the same sentences. Among the 16 heads with the highest kurtoses, we computed the sentence-by-sentence correlation between the vertical-attention scores for each pair of heads; correlated separately for each reasoning trace then averaged (i.e., averaging across numerous correlations with 50-200 samples each). This produced an large correlation (mean $r = .56$). Thus, receiver heads generally attend the same sentences (for reference, the average correlation among any heads is $r = .35$). This convergence across receiver heads is consistent with the existence of sentence importance, which these heads identify.

Attentional narrowing toward particular sentences may be a feature specifically of reasoning models that enhances their performance. Comparing R1-Distill-Qwen-14B (reasoning) and Qwen-14B (base) suggests that the reasoning model's receiver heads will narrow attention toward singular sentences to a greater degree (Section L). Furthermore, ablating receiver heads leads to a greater reduction in accuracy than ablating self-attention heads at random (Section M). Altogether, these findings are consistent with receiver heads and thought-anchor sentences playing key roles in reasoning.

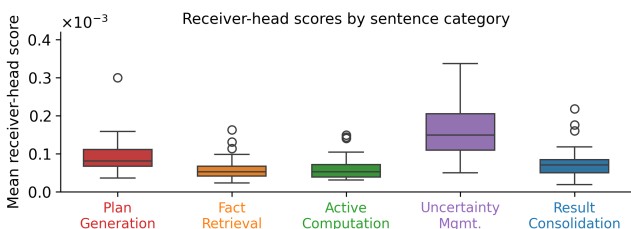

Figure 5: The boxplot shows the average top-32 receiver-head score for each sentence type. The boxes correspond to the interquartile range across different reasoning traces. The effects whereby plan-generation and uncertainty management sentences yield the highest scores ($ps < .001$) remain significant when examining top-16 or top-64 receiver-head scores ($ps < .001$)

## 4.2 LINKS TO COUNTERFACTUAL IMPORTANCE AND SENTENCE TYPES

*Plan generation* and *uncertainty management* sentences consistently receive the most attention via receiver heads (Figure 5), whereas *active computation* sentences receive relatively minimal attention ($ts > 4.0$, $ps < .001$ per paired t-tests comparing the mean receiver-head score for the former two versus the later two categories). These findings demonstrate a parallel between the receiver head findings here and the earlier results on the sentence types yielding the highest counterfactual importance.

## 5 CASE STUDY ON SENTENCE-SENTENCE CAUSAL LINKS

In the previous sections, we focused on how individual sentences influence the final answer. Here, we turn to the finer-grained question of how sentences influence each other within a reasoning trace. Our aim is to estimate directed, sentence-to-sentence causal links: for any pair $(S_i, S_j)$ with $j > i$, how much does altering $S_i$ change the computation carried by $S_j$? We approximate these links by constructing a sentence-level causal graph, using interventions that selectively suppress attention to a given sentence and measuring how this affects the downstream sentence's logits. We first illustrate this approach on our running case study before turning to more systematic analyses.

## 5.1 APPROACH

Our approach examines how suppressing all attention towards a given sentence $S_i$ influences later sentence $S_j$. We define this impact as the KL divergence between logits with/without masking, averaged across a sentence's tokens. We normalize this score by subtracting the latter sentence's average causal effect from all prior sentences. Section N provides pseudocode for generating the causal graph. Suppressing attention is mostly equivalent to omitting a sentence from a CoT, only differing in positional embeddings.

Our approach assumes (i) token logits capture a sentence's semantic content and (ii) masking sentences does not problematically induce out-of-distribution behavior. We evaluated these assumptions by correlating the sentence-sentence scores with those from a sentence-sentence strategy based on our counterfactual resampling method, which assesses how resampling $S_i$ with $T_i$ ($S \not\approx T$) influences the likelihood of $S_j$ appearing. This measure positively correlates with the scores from the masking- and logits-based strategy (section O), suggesting that logits indeed track semantics. We continue with the sentence-masking approach because it requires ∼100x less compute, increasing scalability.

## 5.2 CASE STUDY

We continue our initial case study (Section 2.4), but here, we focus on three local maxima in the sentence-masking graph (Figure 6), which align closely with the sentences implicated as important by receiver-heads (see further details on the case study in Section P):

- **(Sentences: 12 → 43)** After suggesting the answer "20 bits", the model decides to begin verifying it (Sentence 12). Verification leads to a different solution, "19 bits" (Sentence 43). Between these key sentences, most of the intermediate text is performing arithmetic.

- **(Sentences: 44 → 65)** Noticing the discrepancy (Sentence 44), the model decides to check its calculations. It finds that they are correct, and the discrepancy remains (Sentence 65).
- **(Sentences: 12 → 66)** The model realizes that its initial suspicion about leading zeroes (Sentence 12) is justified and states that this is the reason for the discrepancy (Sentence 66).

These connections point to an interpretable scaffold reflecting computations on the pursuit of intermediate results, the execution of self-correction subroutines, and the synthesis of prior statements.

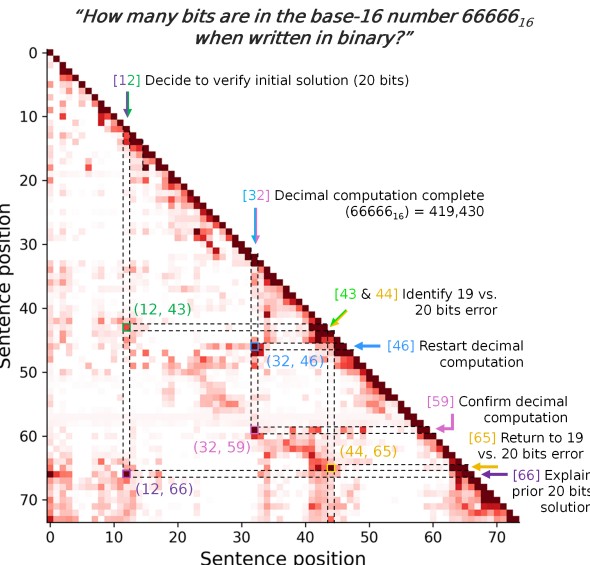

Figure 6: For the correct-answer CoT of Problem #4682, the matrix shows the effect of masking one sentence (x-axis) on a future sentence's logits (y-axis). Darker colors indicate higher values.

## 5.3 OPEN SOURCE INTERFACE

We released an open source interface (anonymous-interface.com) for visualizing reasoning traces and comparing alternative rollouts. We show our proof-of-concept interface in Figure 1C, where important sentences are represented by larger nodes and causal connections between sentences are shown with dashed gray lines. The tool aims to benefit interpretability and unwanted behavior debugging.

## 6 SYSTEMATIC DIFFERENCES IN SENTENCE-SENTENCE CAUSAL LINKS

We next investigated how causal graphs may shed light on general questions about LLM reasoning. Specifically, we ask: How can examining sentence–sentence links shed light on model confidence during reasoning? Relatedly, why do some problem domains like mathematics display stronger uplift in reasoning compared to non-reasoning models? We hypothesize that strong causal links between nearby sentences reflect a coherent logical flow and well-formed plan, so each sentence strongly constrains the next, whereas distant linkages reflect uncertainty and backtracking. Despite occasional long-range connections, we further hypothesize that successful mathematical CoTs are characterized by tight, local causal links between sequential sentences, whereby planning statements sharply structure the CoT by tightly determining what comes next with little variability. Domains related to mathematics may uniquely lend themselves to such firmly structured reasoning, whereas CoTs for other topics (e.g., history or biology) may solve problems by scanning a wider latent space in a less tightly structured fashion.

## 6.1 METHODS

We pivoted to analyzing MMLU problems (Hendrycks et al., 2020), so that we could contrast problem domains. We also switched to Qwen3-30b-a3b, so that we could leverage a serverless LLM provider

that outputs token logits, which allowed scaling up our analysis to thousands of CoTs. We ran Qwen3-30b-a3b in non-reasoning mode on all 15,638 MMLU questions to identify challenging problems where non-reasoning accuracy is under 50% (per answer logits). This corresponds to 3,651 problems, and for 2,492 of these questions, the model answers correctly when using reasoning at least once across ten passes. We computed each correct CoT's causal graph (mean = 90.1 sentences).

We compared graphs on the strength of their causal links at different distances between sentences. We specifically computed the mean attention-suppression effect at distance $k$ for each graph ($m \times m$ sentences) for all $k \leq \frac{m}{2}$. This corresponds to the mean of a matrix's $k$-th subdiagonal. We consider subdiagonals only up to $\frac{m}{2}$ to reduce noise by ensuring that the mean is computed among an adequate number of elements (e.g., the $m$-th subdiagonal would be just the single bottom-leftmost element).

## 6.2 RESULTS

The distance of causal effects tracks question difficulty. Computing correlations within-subject, we find that questions with high average accuracy elicit CoTs with stronger close-range links and weaker long-range links (Figure 7A). In addition, subjects where average accuracy is high overall tend to produce CoTs with stronger close links ($r = .44, p < .001$; Figure 7B) and weaker long links ($r = -.54, p < .001$ Figure 7C). The strongest levels of accuracy were seen in problems requiring mathematical thinking (e.g., mathematics & physics). As hypothesized, these areas also yielded CoTs with stronger close-range connections and weaker long-range connections (two-sample t-test $|t|s > 10, ps < .001$; Figure 7D). Although these analyses do not model *plan generation* and *uncertainty management* sentences directly, they are consistent with a picture in which plan-generation anchors provide the local scaffolding for successful reasoning while uncertainty-management anchors mediate longer-range links that resolve discrepancies, together shaping the overall structure of effective CoTs.

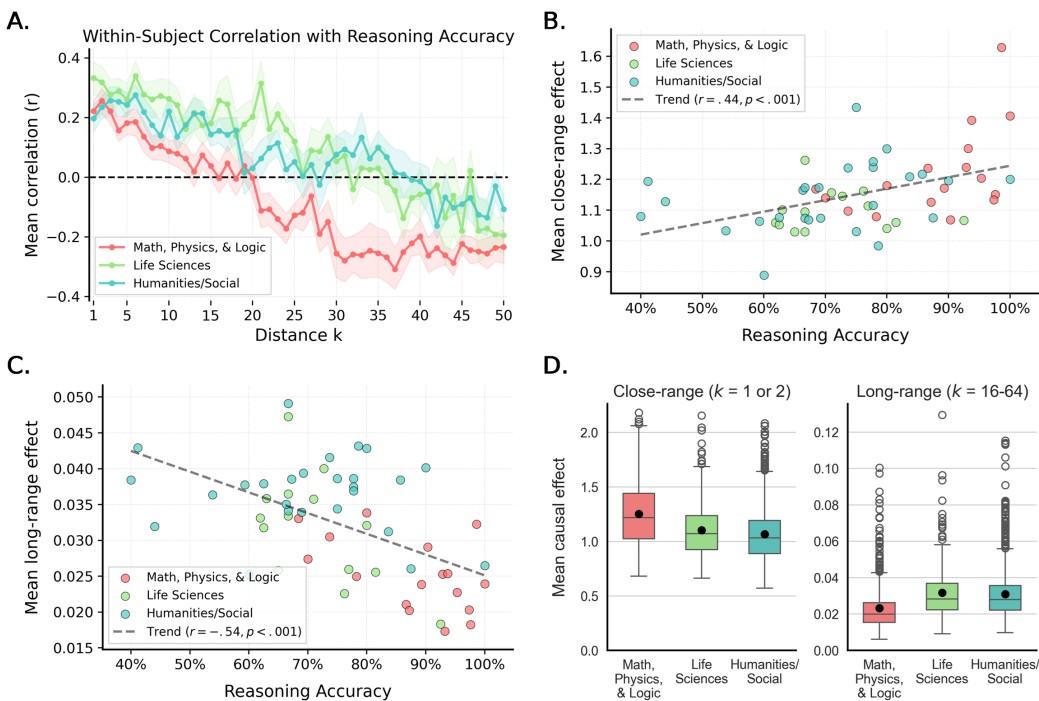

Figure 7: **A.** For each distance $k$, we computed the correlation between a question's average $k$-distance causal effect in one CoT and the question's mean reasoning accuracy across ten CoTs. **B. & C.** Scatterplot shows each subject's average close-range ($k$ = 1-2) and long-range ($k$ = 16-64) was plotted against its average reasoning accuracy. **D.** Box-plots showing the spread of average close-range and long-range causal effects for different question domains; each point represents one CoT, and black circles represent means.

## 7 RELATED WORK

**Reasoning advances and unfaithfulness in LLMs.** CoT reasoning, optimized using reinforcement learning, has driven major capabilities improvements in large language models (Wei et al., 2023; Nye et al., 2021; Reynolds & McDonell, 2021). This reasoning paradigm introduces novel safety challenges. Experiments inducing unfaithful reasoning have led some to raise concerns about the interpretability of CoT text (Lanham et al., 2023b; Chen et al., 2025), although others have argued that CoT text generally is a meaningful representation, particularly for difficult tasks (Korbak et al., 2025). By showing how sentence types, categorized based on their text, differ in their resampling and receiver-head importance, our findings endorse the meaningfulness and interpretability of CoT text.

**Importance of individual steps.** A variety of techniques that can be used for CoT interpretability have been developed, and these likewise suggest that a subset of steps disproportionately drive the final answer – e.g., ROSCOE metrics (Golovneva et al., 2023), gradient-based scores (Wu et al., 2023), and resampling at fork tokens (Bigelow et al., 2024). Complementing these, we provide a more principled framework for understanding how CoTs are constructed around key sentences.

## 8 DISCUSSION AND LIMITATIONS

This work presents initial steps towards a principled decomposition of reasoning traces with a focus on identifying thought anchors: sentences with outsized importance on the model's final response, specific future sentences, and downstream reasoning trajectory. We have also begun unpacking the attentional mechanisms associated with these important sentences. We expect that understanding thought anchors will be critical for interpreting reasoning models and ensuring their safety.

While some research raises concerns that CoT text can be unfaithful to the model's underlying computation (Lanham et al., 2023b; Chen et al., 2025), our results show CoT text is mechanistically relevant and interpretable. For example, sentences categorized as *plan generation* and *uncertainty management* consistently exhibit higher counterfactual importance in our resampling analyses and receive more focused attention from receiver heads. This demonstrates a link between what a sentence says and its functional role in the computation, and this type of correspondence supports arguments on the value of CoT legibility (Korbak et al., 2025).

A primary limitation of our resampling approach is its computational cost. For a CoT of 150 sentences and 4000 tokens, resampling each sentence 100 times corresponds to 20M output tokens. This cost precludes its usage for real-time monitoring, but it remains feasible for intensive analyses of specific questions of interest – e.g., understanding CoTs in safety-relevant scenarios, like cases of LLM blackmailing or reward hacking. We resampled CoTs 100 times per sentence to achieve fairly precise estimates (in terms of final-answer accuracy, 95% CI corresponds to at worst $\pm 10\%$). However, for analyses that average effects across many CoTs, fewer resamples for any one CoT would suffice. Future work could develop adaptive resampling strategies that allocate more computational budget to potentially pivotal moments in the trace, maximizing precision while minimizing cost.

Further work is needed to evaluate the generalizability of our findings across model capabilities and question types. More advanced models may display improved error correction abilities, lowering the frequency of sudden drops in accuracy following a sentence. Such models may also be more aggressively trained to minimize CoT length, which could increase average importance. Problem difficulty also influences reasoning, as we show in Section 6. Extremely difficult problems might contain numerous points for subtle errors that could be difficult to correct; in this case, correct-CoT sentences may be mostly low counterfactual importance, while incorrect CoTs could contain large downward spikes following any error. Research remains necessary to uncover the landscape of reasoning behavior, but we expect our methods will still apply to larger models and other problems.

We view this as preliminary work. Our analyses require refinement to grapple with how downstream sentences may be overdetermined by different possible trajectories or independent sufficient causes. Our receiver-head analyses are confounded by a sentence's position in the reasoning trace (see Section Q). Despite these limitations, we believe that we have demonstrated that our metrics are an advance on prior work, interrupting models and forcing final answers. The surprising degree of shared structure we have found across our three methods illustrates the potential value of future work in this area and points to the possibility of more powerful interpretability techniques to come.

## 9 REPRODUCIBILITY STATEMENT

To ensure the reproducibility of our work, we provide comprehensive implementation details, code, and experimental specifications. Our code is publicly available at `https://anonymous.4open.science/r/thought-anchors-2CB0`, which includes all scripts for black-box resampling, receiver head analysis, and attention suppression experiments. We have also released two Python packages to aid in conducting these analyses: The first package helps with CoT prefilling and caching API responses (`https://anonymous.4open.science/r/rollouts-6C4D/`), and the second package helps with properly splitting CoTs into sentences while respecting standard tokenization procedures (`https://anonymous.4open.science/r/Sentences-1148`). We also provide an interactive visualization tool at `anonymous-interface.com` for exploring reasoning traces and sentence-level causal dependencies.

The complete prompt used for sentence taxonomy labeling is provided in Section D, including detailed instructions for function tags and dependency annotations. Our experimental setup uses DeepSeek R1-Distill-Qwen-14B (48 layers) with temperature 0.6 and top-p 0.95, tested on the MATH dataset (Hendrycks et al., 2020) focusing on problems with 25-75% solution rates. We specify exact hyperparameters including 100 rollouts per sentence for counterfactual resampling, cosine similarity threshold of 0.8 (median value) using `all-MiniLM-L6-v2` embeddings (Section F), and identification of receiver heads via kurtosis scores of attention distributions. The sentence-sentence causal masking methodology is fully detailed in Section 5, with validation through correlation with resampling-based measures (Section O). For MMLU experiments in Section 6, we used Qwen3-30b-a3b on 2,492 problems where non-reasoning accuracy is below 50%, computing causal graphs for correct CoTs. Additional reproducibility details include: full case study transcript (Section A), sentence category distributions (Section C), receiver head ablation procedures with 128/256/512 heads (Section M), and cross-model validation on R1-Distill-Llama-8B (Section B). All models used are publicly available, and we provide pseudocode for the sentence-to-sentence importance calculation in Section O.

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

## A   CASE STUDY DETAILS

### A.1   CASE STUDY TRANSCRIPT

Below is a transcript of the model's response used for the case study (Section 5.2). The first 74 sentences following <think> are shown. After this point, the model converged on 100% resampling accuracy for all subsequent sentences.

0. (problem_setup): Okay, so I have this problem where I need to find out how many bits the base-16 number $66666_{16}$ has when it's converted to base-2.

1. (uncertainty_management): Hmm, let's see.

2. (fact_retrieval): I remember that each hexadecimal digit corresponds to exactly 4 binary digits, or bits.

3. (plan_generation): So, maybe I can just figure out how many hexadecimal digits there are and multiply by 4 to get the number of bits.

4. (uncertainty_management): Let me check that.

5. (fact_retrieval): First, the number given is 66666 in base 16.

6. (active_computation): Let me count the digits: 6, 6, 6, 6, 6.

7. (result_consolidation): That's five digits in total.

8. (active_computation): So, if each digit is 4 bits, then 5 digits would be 5×4=20 bits.

9. (uncertainty_management): Wait, is that always the case?

10. (fact_retrieval): I think so because each hex digit maps directly to 4 bits without any overlap or sharing.

11. (result_consolidation): So, $66666_{16}$ would be 20 bits in binary.

12. (uncertainty_management): But wait, I should make sure that this is correct by actually converting the number or at least checking if there's any leading zero that might affect the bit count.

13. (plan_generation): Alternatively, maybe I can calculate the value of $66666_{16}$ in decimal and then find out how many bits that number would require.

14. (plan_generation): Let's try that approach to cross-verify.

15. (fact_retrieval): So, to convert $66666_{16}$ to decimal, I can expand it as follows:

16. (fact_retrieval): Each digit represents a power of 16.

17. (fact_retrieval): Starting from the right, the digits are $16^0, 16^1, 16^2, 16^3, 16^4$.

18. (fact_retrieval): So, the number is:

19. (active_computation): $6×16^4+6×16^3+6×16^2+6×16^1+6×16^0$

20. (plan_generation): Let me compute each term:

21. (fact_retrieval): First, $16^4$ is 16×16×16×16.

22. (active_computation): $16^2$ is 256, so $16^3$ is 256×16=4096, and $16^4$ is 4096×16=65536.

23. (active_computation): So, 6×65536=393216

24. (active_computation): Next, $16^3$ is 4096, so 6×4096=24576

25. (active_computation): Then, $16^2$ is 256, so 6×256=1536

26. (active_computation): $16^1$ is 16, so 6×16=96

27. (active_computation): And $16^0$ is 1, so 6×1=6

28. (result_consolidation): Now, adding all these up:

29. (active_computation): 393216+24576=417,792

30. (active_computation): 417,792+1536=419,328

31. (active_computation): 419,328+96=419,424

32. (`active_computation`): 419,424+6=419,430

33. (`result_consolidation`): So, $66666_{16}$ is equal to 419,430 in decimal.

34. (`fact_retrieval`): Now, to find out how many bits this number requires, I can use the formula for the number of bits needed to represent a number N in binary, which is $floor(log_2(N)) + 1$.

35. (`plan_generation`): So, let's compute $log_2(419, 430)$.

36. (`uncertainty_management`): Hmm, I don't have a calculator here, but I can estimate it.

37. (`fact_retrieval`): I know that $2^{18}$ =262,144

38. (`fact_retrieval`): $2^{19}$ =524,288

39. (`result_consolidation`): So, 419,430 is between $2^{18}$ and $2^{19}$.

40. (`result_consolidation`): Therefore, $log_2(419, 430)$ is between 18 and 19.

41. (`active_computation`): The floor of that would be 18, so the number of bits is 18+1=19 bits.

42. (`uncertainty_management`): Wait, but earlier I thought it would be 20 bits because it's 5 hex digits each being 4 bits.

43. (`result_consolidation`): Now, according to this, it's 19 bits.

44. (`uncertainty_management`): There's a discrepancy here.

45. (`uncertainty_management`): I must have made a mistake somewhere.

46. (`self_checking`): Wait, let me check my calculations again.

47. (`uncertainty_management`): Maybe I messed up the decimal conversion.

48. (`fact_retrieval`): Wait, 66666 in hex.

49. (`plan_generation`): Let me convert it step by step again.

50. (`active_computation`): $6 \times 16^4$: 6×65536=393216

51. (`active_computation`): $6 \times 16^3$: 6×4096=24576

52. (`active_computation`): $6 \times 16^2$: 6×256=1536

53. (`active_computation`): $6 \times 16^1$: 6×16=96

54. (`active_computation`): $6 \times 16^0$: 6×1=6

55. (`active_computation`): Adding them up: 393216+24576=417,792

56. (`active_computation`): 417,792+1536=419,328

57. (`active_computation`): 419,328+96=419,424

58. (`active_computation`): 419,424+6=419,430.

59. (`result_consolidation`): That seems correct.

60. (`result_consolidation`): So, 419,430 in decimal.

61. (`fact_retrieval`): Now, let's check $2^{18}$ is 262,144, $2^{19}$ is 524,288.

62. (`result_consolidation`): So, 419,430 is less than 524,288, so $log_2(419, 430)$ is less than 19.

63. (`active_computation`): So, $floor(log_2(419, 430))$ is 18, so number of bits is 19.

64. (`uncertainty_management`): But wait, when I thought about each hex digit being 4 bits, 5 hex digits would be 20 bits.

65. (`uncertainty_management`): So, why is there a difference?

66. (`plan_generation`): Ah, perhaps because leading zeros are not counted.

67. (`fact_retrieval`): So, when converting from hex to binary, each digit is 4 bits, but if the leading hex digits are such that when converted to binary, they don't add any leading zeros beyond the 4 bits per digit.

68. (`uncertainty_management`): Wait, but 66666 in hex is 5 digits, so 5×4=20 bits.

69. (`result_consolidation`): But according to the decimal conversion, it's 19 bits.

70. (`result_consolidation`): That suggests that when converted to binary, it's 19 bits, not 20.

71. (`uncertainty_management`): So, which is correct?

72. (`plan_generation`): Wait, perhaps when converting $66666_{16}$ to binary, it's exactly 20 bits, but the leading bits are zeros, so when counting the number of bits, we don't count leading zeros.

73. (`fact_retrieval`): Wait, but in the problem, it's just asking how many bits it has when written in base 2.

### A.2 CASE STUDY: SENTENCE 13 ALTERNATIVES

Sentence 13 (*"Alternatively, maybe I can calculate the value of $66666_{16}$ in decimal and then find out how many bits that number would require."*) was found to have the highest counterfactual importance among any sentence in the response (see Figure 2A). Five alternative possible sentences resampled at the sentence 13 position, marked by whether they eventually led to a correct or incorrect answer, are as follows:

1. ($\times$) Let me think. The first digit is 6, which in binary is 0110.
2. ($\checkmark$) Let me try converting the number to decimal first...
3. ($\times$) Let me try converting the first few digits to binary to see how it goes.
4. ($\times$) Let me think about the conversion process.
5. ($\checkmark$) Let me try converting the number step by step.

## B GENERALIZING TO AN ALTERNATIVE REASONING MODEL

### B.1 MEASURING COUNTERFACTUAL INFLUENCE

To assess the generalizability of our counterfactual importance findings, we replicated our resampling methodology on R1-Distill-Llama-8B, applying the same experimental parameters (e.g., temperature = 0.6 and top-p = 0.95) used for R1-Distill-Qwen-14B. We collected 100 rollouts for 20 correct and 20 incorrect base solutions using the identical question set described in Section 2.

The resampling accuracy trajectories for R1-Distill-Llama-8B (Figure 8) demonstrate patterns that are similar to those observed in R1-Distill-Qwen-14B (Figure 2). Specifically, we observe similar characteristic accuracy fluctuations throughout the reasoning traces, with notable spikes and dips occurring at sentences corresponding to critical reasoning transitions.

Figure 9 shows that R1-Distill-Llama-8B exhibits similar sentence category effects whereby *plan generation* and *uncertainty management* sentences demonstrate higher counterfactual importance compared to *active computation* and *fact retrieval* sentences (see Figure 3 for R1-Distill-Qwen-14B).

This cross-model validation supports our claim that reasoning traces are structured around high-level organizational sentences rather than low-level computational steps. The consistency of counterfactual importance patterns suggests that our sentence-level attribution framework captures fundamental properties of chain-of-thought reasoning that generalize beyond specific model implementations.

### B.2 ATTENTION AGGREGATION

R1-Distill-Llama-8B displayed receiver-head patterns largely consistent with those of R1-Distill-Qwen-14B. The histogram of attention heads' vertical-attention scores displays a right tail, indicating that some attention heads tend to particularly focus attention on a subset of sentences (Figure 11A). Interestingly, the R1-Distill-Qwen-14B receiver-heads tended to be more frequent in later layers (see below, Figure 20), which was not evident in R1-Distill-Llama-8B (Figure 10).

The R1-Distill-Qwen-14B and R1-Distill-Llama-8B receiver heads displayed consistent patterns related to sentence types, such that *plan generation*, *uncertainty management*, and *self checking*

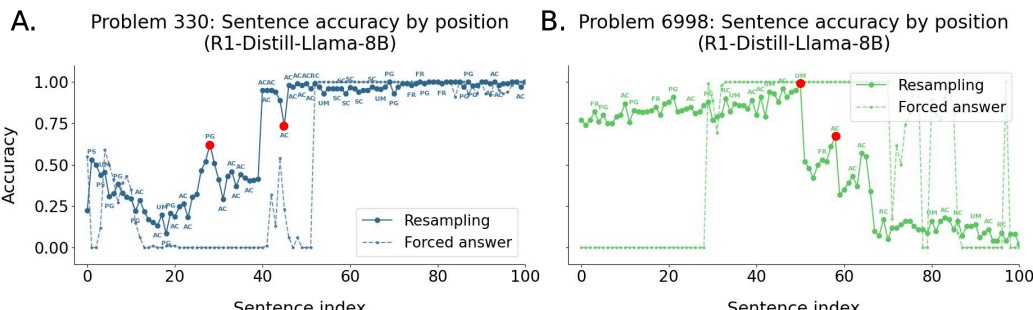

Figure 8: Accuracy over 100 rollouts at each sentence for (**A**) one correct and (**B**) one incorrect base solution for R1-Distill-Llama-8B. Red dots mark significant spikes or dips. Local minima and maxima sentences are annotated with category initials. Our analyses focus on the counterfactual KL-divergence between sentences, but resampling accuracy is visualized here as it is more intuitive.

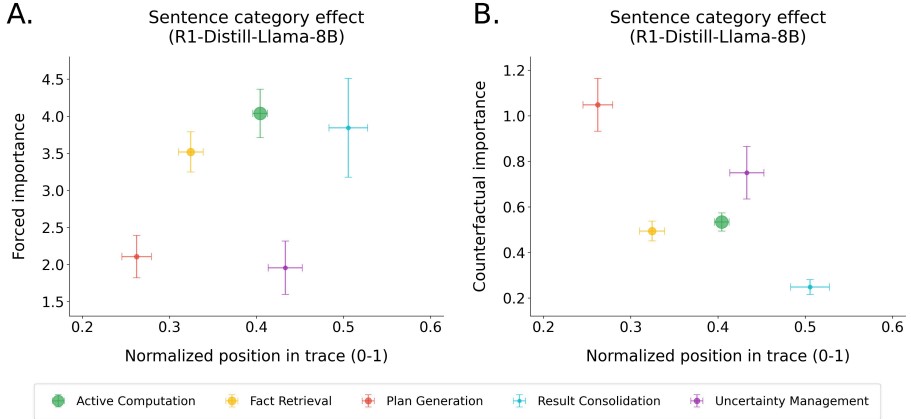

Figure 9: The mean of each sentence category for (**A**) forced-answer importance and (**B**) counterfactual importance for R1-Distill-Llama-8B, per the resampling method, plotted against the sentence category's mean position in the reasoning trace. Only the 5 most common sentence types are shown.

sentences received heightened attention; although visually, the differences to *fact retrieval* and *active computation* may be less prominent, paired t-tests (paired with respect to a given response) showed that *plan generation* and *uncertainty management* always significantly surpassed *fact retrieval* and *active computation* (four paired t-tests: $ps \leq .01$).

No R1-Distill-Llama-8B results are provided for the attention suppression analysis, as that method was principally used for the case study, and no new case study was performed for R1-Distill-Llama-8B.

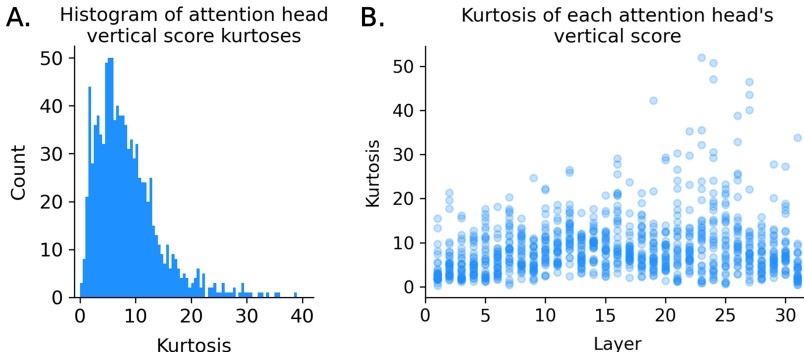

Figure 10: The plots here show the vertical-attention score patterns associated with the R1-Distill-Llama-8B data. **A.** This histogram shows the kurtosis values across all attention heads, median across all reasoning traces; parallels Figure 4 based on the R1-Qwen-14B data. **B.** This scatterplot shows the kurtosis of each head's vertical-attention score, organized by layer. Figure 20 is the R1-Distill-Qwen-14B version of this figure, which showed an upward trend into later layers that is not evident here.

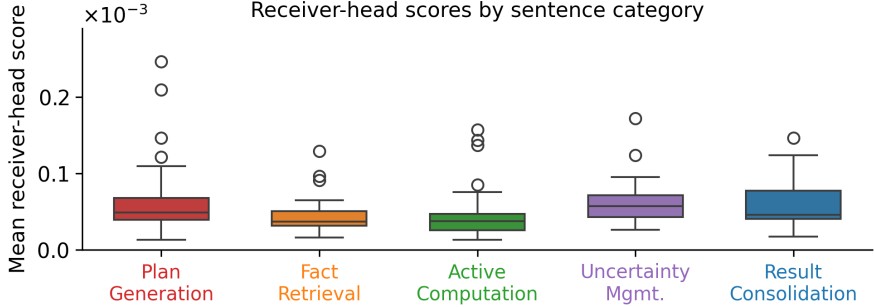

Figure 11: Based on the R1-Distill-Llama-8B data, the boxplot shows the average top-64 receiver-head score for each sentence type. The boxes correspond to the interquartile range across different reasoning traces. Figure 5 is the R1-Distill-Qwen-14B version of this figure; note that for the R1-Distill-Qwen-14B figure, the top-32 heads were used. We found that for Llama 8B, examining the top-64 heads yielded more pronounced differences, although the sentence types with the highest scores remain the same.

## C  SENTENCE TAXONOMY

Building on top of the framework presented by (Venhoff et al., 2025), we developed a taxonomy consisting of eight distinct sentence categories that capture reasoning functions in mathematical problem-solving. Each category represents a specific cognitive operation. The functions and examples for each category are given in Table 1 and Table 2. Notably, the *uncertainty management* category includes backtracking sentences.

Table 1: Sentence taxonomy with reasoning functions in problem-solving

| Category | Function | Examples |
|---|---|---|
| **Problem Setup** | Parsing or rephrasing the problem (e.g., initial reading) | *I need to find the area of a circle with radius 5 cm.* |
| **Plan Generation** | Stating or deciding on a plan of action, meta-reasoning | *I'll solve this by applying the area formula.* |
| **Fact Retrieval** | Recalling facts, formulas, problem details without computation | *The formula for the area of a circle is $A = \pi r^2$.* |
| **Active Computation** | Algebra, calculations, or other manipulations toward the answer | *Substituting $r = 5$: $A = \pi \times 5^2 = 25\pi$.* |
| **Uncertainty Management** | Expressing confusion, re-evaluating, including backtracking | *Wait, I made a mistake earlier. Let me reconsider...* |
| **Result Consolidation** | Aggregating intermediate results, summarizing, or preparing | *So the area is $25\pi$ square cm which is approximately...* |
| **Self Checking** | Verifying previous steps, checking calculations, and re-confirmations | *Let me verify: $\pi r^2 = \pi \times 5^2 = 25\pi$. Correct.* |
| **Final Answer Emission** | Explicitly stating the final answer | *Therefore, the answer is...* |

The distribution of categories across our dataset as shown in Figure 12 reveals that *active computation* constitutes the largest proportion (32.7%), followed by *fact retrieval* (20.1%), *plan generation* (15.5%), and *uncertainty management* (14.0%). The sequential structure of reasoning is reflected in the rarity and positioning of *problem setup* (2.4%), which typically occurs at the beginning, and *final answer emission* (0.7%), which predominantly appears toward the end of the reasoning process.

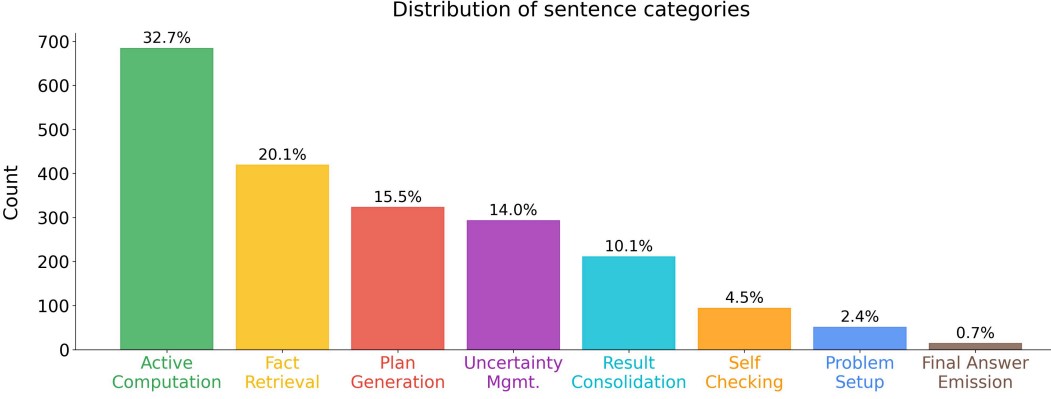

Figure 12: Counts and frequencies of taxonomic sentence categories in our dataset.

Table 2: Ten randomly selected examples for each category in our functional taxonomy. Examples are drawn from model reasoning traces on MATH problems.

| # | Example Sentence |
|---|---|
| **Problem Setup** (n=124) | |

Table 2 – continued from previous page

| # | Example Sentence |
|---|---|
| 1 | The question is asking how many different ways we can assign 0s and 1s to the bottom row such that the number in the ver... |
| 2 | So, each of these four small triangles is a quarter of the size of the large one, and they are arranged such that one is... |
| 3 | The answer should be to the nearest hundredth of a percent. |
| 4 | Compute $3(1 + 3(1 + 3(1 + 3(1 + 3(1 + 3(1 + 3(1 + 3(1 + 3)))))))))$ |
| 5 | But wait, the problem is asking for the coefficient in the entire expression \( (a + b)^4 \left( c + \frac1c \right)... |
| 6 | Okay, so I've got this problem where a mother wants to arrange her plates around a circular table, and she doesn't want ... |
| 7 | So, it's a triangle with 11 rows, right? |
| 8 | So, the large triangle is divided into four small ones, each at the corners, and the central area is another shape, but ... |
| 9 | Wait, but in an equilateral triangle, if you connect the midpoints, you actually create four smaller equilateral triangl... |
| 10 | Wait, the problem is asking for the coefficient of \(a^2b^2\). |

**Plan Generation** (n=761)

| # | |
|---|---|
| 1 | Let's check: |
| 2 | Since P is a polynomial whose roots are r_j + 1/r_j, maybe I can express P in terms of f(x). |
| 3 | Let me try expanding both parts separately. |
| 4 | Suppose that two different irrational roots have the same conjugate. |
| 5 | Let me see how that progresses over 12 months. |
| 6 | Then, we can model this as a linear algebra problem over the field GF(3), where we're looking for the number of solution... |
| 7 | If I can figure out what those coefficients are modulo 3, I can then determine how the initial 0s and 1s in the bottom r... |
| 8 | When they're nailed together, the area where they overlap would be covered by both boards, meaning that when you paint t... |
| 9 | Now, I need to find P(1)/P(-1). |
| 10 | Similarly, f($\delta$) = $\delta^{2007}$ +17$\delta^{2006}$ +1. |

**Fact Retrieval** (n=1470)

| # | |
|---|---|
| 1 | Wait, but if r is 2 feet, and h is 3 feet, then (r - h) would be negative, which is (2 - 3) = -1. |
| 2 | However, the problem states that \( r \), \( s \), and \( t \) are distinct, so \( c = 1 \) is a possible solution, but ... |
| 3 | 3. Rotation by 240° clockwise: this is the inverse of the 120° rotation, so it would be (1 2 4 3). |
| 4 | $\gamma$ is a primitive 3rd root of unity, so $\gamma$^3 = 1, and $\gamma$^2 = $\gamma$^-1. |
| 5 | Given y = x + 1/x, then as I mentioned before, x satisfies $x^2$ - yx + 1 = 0. |
| 6 | So, sin($\theta$) = 0 when $\theta$ = n$\pi$, where n is any integer. |
| 7 | So, starting from the bottom side, for example, from the bottom-left corner, moving 3 units to the right is the first po... |
| 8 | Points B and A are at (0,2) and (12,10), respectively. |
| 9 | So, these two equations relate r and 1/r. |
| 10 | I think that's correct because each of the four variables a1, a2, a10, a11 can be 0 or 1, giving 16 possibilities. |

**Active Computation** (n=1902)

| # | |
|---|---|
| 1 | Simplify numerator and denominator: |
| 2 | Wait, but when k = 319, x = 1/(319$\pi$) $\approx$ 0.0009965, which is still less than 0.001, so it's included. |
| 3 | $3 * 500 = 1500$ |

Table 2 – continued from previous page

| # | Example Sentence |
|---|---|
| 4 | 2.296... $\div$ 3 $\approx$ 0.765... |
| 5 | Compute each term: |
| 6 | Therefore, the total boundary length is $12 + 6\pi$ units. |
| 7 | $1.005^5 \approx 1.0201503906 * 1.005 \approx 1.0252518971$ |
| 8 | Left side: \( (z - 0)(z - s)(z + s) = z(z^2 - s^2) = z^3 - s^2 z \) |
| 9 | 10/24 = 120/288 |
| 10 | Then, when upright, $V = 4\pi H$. |

**Result Consolidation** (n=879)

| | |
|---|---|
| 1 | So, the minimal degree is 1970. |
| 2 | Therefore, the only possible c's are the cube roots of unity, which are three in number: $1, \omega, \omega^2$. |
| 3 | y=2 $\pm$ 19/81, so y$\approx$2.2345679 or y$\approx$1.7654321. |
| 4 | Thus, y is either 2 + 19/81 or 2 - 19/81, which are 181/81 and 143/81, respectively. |
| 5 | Since each root requires its conjugate, and each pair contributes 2 to the degree, and there are 1000 such roots, each n... |
| 6 | Alternatively, I can write this as 143/(3^9), since 3^9 is 19,683. |
| 7 | Thus, I'm confident that p(5) = -5/12. |
| 8 | So, to find x such that f(x)=186, we can write 186 = 3^n $*$ f(y), where y is in [1,3], and f(y) is in [0,1]. |
| 9 | Therefore, the final result is 36,084. |
| 10 | So, my calculation up to Level 9 as 36084 seems correct. |

**Self Checking** (n=392)

| | |
|---|---|
| 1 | / (5!2!1!) = 8! |
| 2 | Correct. So, yes, 187 - 98 = 89. |
| 3 | But since the plates are arranged in a circle, we use (n-1)! |
| 4 | Yes, that's correct. |
| 5 | As I thought, 289 is 17^2, 259 is 7$*$37. |
| 6 | Wait, let me make sure: the number of integers from a to b inclusive is b - a + 1. |
| 7 | 48+50 is 98, plus 89 is 187. |
| 8 | 4. Then I noticed that every 3 functions, it cycles: f1, x, f, f1, x, f, etc. |
| 9 | So, in \((a + b)^4\), the coefficient of \(a^2b^2\) is indeed 6. |
| 10 | Layer 8: 1 + 3$*$3280 = 1 + 9840 = 9841 |

**Uncertainty Management** (n=859)

| | |
|---|---|
| 1 | So, I must have been wrong with that formula. |
| 2 | But wait, that would mean that the leading zero can be dropped, making it 19 bits. |
| 3 | Wait, but let me verify this because I might be making a mistake. |
| 4 | Wait, but let me confirm the count for S=0 and S=3. |
| 5 | Wait, perhaps I made a mistake. |
| 6 | Wait, but let me think again: when I subtract 3183 - 319, that's 2864, and adding 1 gives 2865. |
| 7 | Wait, no, maybe I'm getting confused. |
| 8 | Let me do it properly. |
| 9 | Wait, perhaps the issue is that when converting from hex to binary directly, the leading zeros in the binary digits migh... |
| 10 | But let me think more carefully. |

**Final Answer Emission** (n=87)

| | |
|---|---|
| 1 | So, the final answer is -289/259. |
| 2 | Therefore, there are 89 more 4's than 8's. |
| 3 | $**$Final Answer$**$ The number of initial distributions is \boxed1280. |
| 4 | So, the number of x-intercepts is 2865. |

| # | Example Sentence |
|---|---|
| | Table 2 – continued from previous page |
| 5 | **Final Answer** The smallest $\( x \)$ for which $\( f(x) = f(2001) \)$ is $\(\boxed\dfrac14319683\)$. |
| 6 | So, f_1993(3) = -1/2. |
| 7 | Therefore, the coefficient of $\( a^2b^2 \)$ in the entire expression is 384. |
| 8 | Therefore, the area of the unpainted region on the four-inch board is $12\sqrt{3}$ square inches. |
| 9 | So, the number of distinguishable large equilateral triangles is 336. |
| 10 | Therefore, p(5) = 24/5. |

## D  PROMPT INFORMATION

We used the following prompt with OpenAI GPT-4o (April-May, 2025) to annotate each sentence:

```
You are an expert in interpreting how LLMs solve math problems
using multi-step reasoning.  Your task is to analyze a
chain-of-thought reasoning trace, broken into discrete text
sentences, and label each sentence with:

1.  **function_tags**:  One or more labels that describe what this
sentence is *doing* functionally in the reasoning process.
2.  **depends_on**:  A list of earlier sentence indices that this
sentence directly depends on, e.g., uses information, results, or
logic introduced in earlier sentences.

This annotation will be used to build a dependency graph and
perform causal analysis, so please be precise and conservative:
only mark a sentence as dependent on another if its reasoning
clearly uses a previous sentence's result or idea.

Function Tags:

1.  problem_setup:  Parsing or rephrasing the problem (initial
reading or comprehension).
2.  plan_generation:  Stating or deciding on a plan of action
(often meta-reasoning).
3.  fact_retrieval:  Recalling facts, formulas, problem details
(without immediate computation).
4.  active_computation:  Performing algebra, calculations,
manipulations toward the answer.
5.  result_consolidation:  Aggregating intermediate results,
summarizing, or preparing final answer.
6.  uncertainty_management:  Expressing confusion, re-evaluating,
proposing alternative plans (includes backtracking).
7.  final_answer_emission:  Explicit statement of the final boxed
answer or earlier sentences that contain the final answer.
8.  self_checking:  Verifying previous steps, checking
calculations, and re-confirmations.
9.  unknown:  Use only if the sentence does not fit any of the
above tags or is purely stylistic or semantic.

Dependencies:

For each sentence, include a list of earlier sentence indices that
```

```
the reasoning in this sentence *uses*.  For example:
- If sentence 9 performs a computation based on a plan in sentence
4 and a recalled rule in sentence 5, then depends_on:  [4, 5]
- If sentence 24 plugs in a final answer to verify correctness
from sentence 23, then depends_on:  [23]
- If there's no clear dependency use an empty list:  []
- If sentence 13 performs a computation based on information in
sentence 11, which in turn uses information from sentence 7, then
depends_on:  [11, 7]

Important Notes:
- Make sure to include all dependencies for each sentence.
- Include both long-range and short-range dependencies.
- Do NOT forget about long-range dependencies.
- Try to be as comprehensive as possible.
- Make sure there is a path from earlier sentences to the final
answer.
Output Format:

Return a dictionary with one entry per sentence, where each entry
has:
- the sentence index (as the key, converted to a string),
- a dictionary with:
  - "function_tags":  list of tag strings
  - "depends_on":  list of sentence indices, converted to strings

Here is the expected format:
{
  "1":  {
   "function_tags":  ["problem_setup"],
   "depends_on":  [""]
  },
  "4":  {
   "function_tags":  ["plan_generation"],
   "depends_on":  ["3"]
  },
  "5":  {
   "function_tags":  ["fact_retrieval"],
   "depends_on":  []
  },
  "9":  {
   "function_tags":  ["active_computation"],
   "depends_on":  ["4", "5"]
  },
  "24":  {
   "function_tags":  ["uncertainty_management"],
   "depends_on":  ["23"]
  },
  "32":  {
   "function_tags":  ["final_answer_emission"],
   "depends_on":  ["9", "30", "32"]
  },
}

Here is the math problem:
<PROBLEM>

Here is the full chain-of-thought, broken into sentences:
<SENTENCES>
```

```
Now label each sentence with function tags and dependencies.
```

## E    SENTENCE CATEGORY PROBING

We trained a linear classifier to identify sentence categories based on activations. We employed a multinomial logistic regression with L2 regularization ($C = 1.0$) on the residual stream activity from layer $47$ (last layer) of R1-Distill-Qwen-14B. For evaluating accuracy, we implemented a group-5-fold cross-validation that ensured examples from the same problem response remained in either the training or testing set to prevent data leakage. We averaged the residual stream activity across tokens to create sentence-level representations, whose dimensions were then standardized. To address class imbalance in the training data, we employed balanced class weights. The model demonstrated strong discriminative power across all reasoning categories, achieving a macro-F1 score of $0.71$. The confusion matrix presented in Figure 13 reveals high classification accuracies for categories such as *active computation* ($0.74$), *uncertainty management* ($0.79$), and *problem setup* ($0.83$), while showing some confusion between functionally related categories.

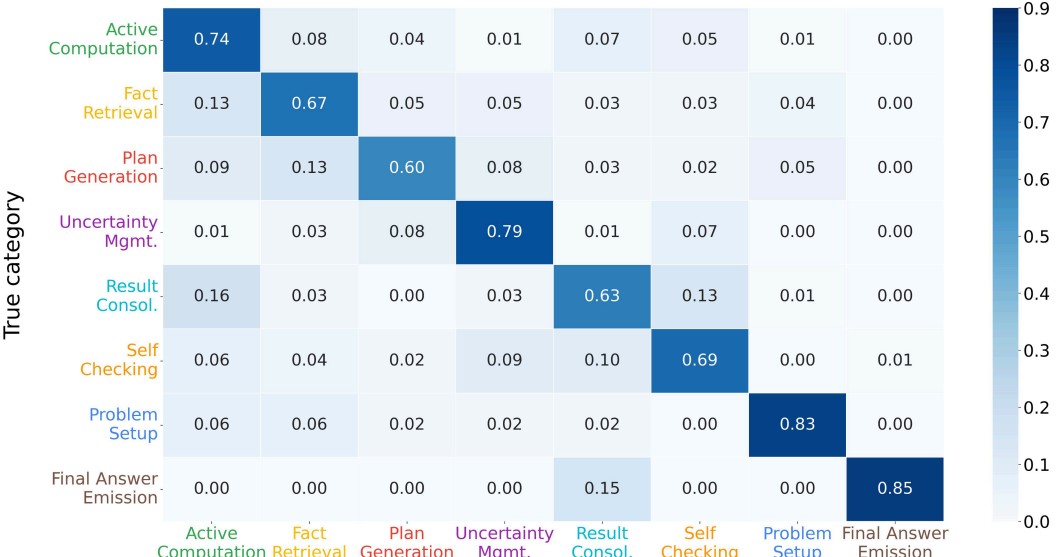

Figure 13: Confusion matrix showing the sentence category classification performance of a logistic regression probe trained on activations from layer 47 of the R1-Distill-Qwen-14B model. Values represent the proportion of examples from each true category (rows) classified as each predicted category (columns). Diagonal elements indicate correct classifications.

## F    EMBEDDINGS MODEL AND COUNTERFACTUAL IMPORTANCE RESULTS ACROSS SIMILARITY THRESHOLDS

We used `all-MiniLM-L6-v2` with a maximum sequence length of $256$ tokens and a hidden dimension of $384$ as our sentence embeddings model from the `sentence-transformers` (Reimers & Gurevych, 2019) library. We picked a cosine similarity threshold of $0.8$, which is the median similarity value between all sentence removed (i.e., original sentence) and sentence resampled pairs in our dataset.

The effects reported in Figure 3, whereby *plan generation* and *uncertainty management* display the highest levels of counterfactual importance, also emerges when the cosine similarity threshold for $T_i \not\approx S_i$ is set at $0.5$ or $0.9$ (Figure 14).

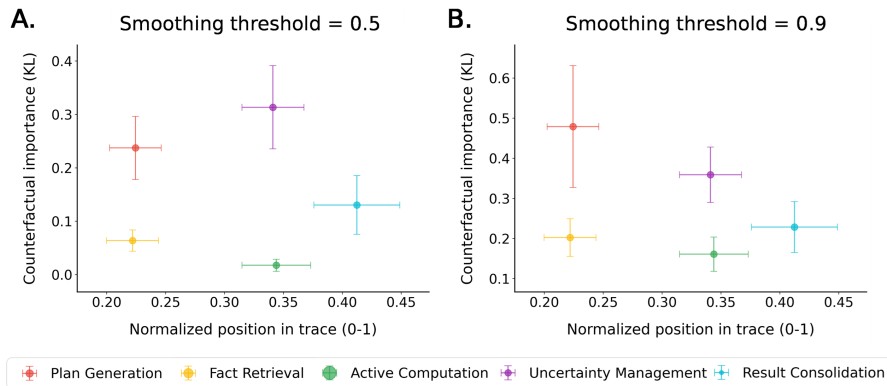

Figure 14: This is a variant of Figure 3, now performed a cosine similarity threshold of using either **(A)** 0.5 or **(B)** 0.9

## G   EVALUATING IMPORTANCE (KL) WHILE SMOOTHING

The identified link between sentence's category and its forced-answer or counterfactual importances were also measured while smoothing the final-answer distribution associated with each sentence. Smoothing was performed when computing the KL divergence between the two distribution and constitutes replacing the $\epsilon = 10^{-9}$ term (originally used to avoid division by zero) with $\alpha = 1.0$ (Laplace smoothing) or $\alpha = 0.5$ (smoothing with Jeffrey's prior).

Let $p(A'_{S_i})$ and $p(A_{S_i})$ be the empirical distributions over a set of $K$ possible final answers, $\mathcal{A}$, derived from $N$ rollouts (e.g., $N = 100$). Let $C'_{S_i}(a)$ and $C_{S_i}(a)$ be the observed counts for a specific answer $a \in \mathcal{A}$ in the intervention and base conditions, respectively, such that $\sum_{a \in \mathcal{A}} C'_{S_i}(a) = N$. Additive smoothing with a parameter $\alpha$ is applied to derive smoothed probabilities, $p_\alpha$ and $q_\alpha$, from these counts:

$$p_\alpha(a) = \frac{C'_{S_i}(a) + \alpha}{N + K\alpha} \quad \text{and} \quad q_\alpha(a) = \frac{C_{S_i}(a) + \alpha}{N + K\alpha}$$

The smoothed KL divergence, $D_{\text{KL}}^\alpha$, is then computed using these non-zero probabilities:

$$D_{\text{KL}}^\alpha[p(A'_{S_i})||p(A_{S_i})] = \sum_{a \in \mathcal{A}} p_\alpha(a) \log \left( \frac{p_\alpha(a)}{q_\alpha(a)} \right)$$

This method replaces the use of a small $\epsilon$ floor. The smoothing parameters used are $\alpha = 1.0$ (Laplace smoothing) and $\alpha = 0.5$ (Jeffreys prior).

With either level of smoothing, the same patterns linking importance and sentence category emerge as initially reported without smoothing (Figure 3). Specifically, *active computation* sentences yield higher forced answer importance than *plan generation* and *uncertainty management*, but the reverse is true when examining counterfactual importance based on the resampling method (Figure 15).

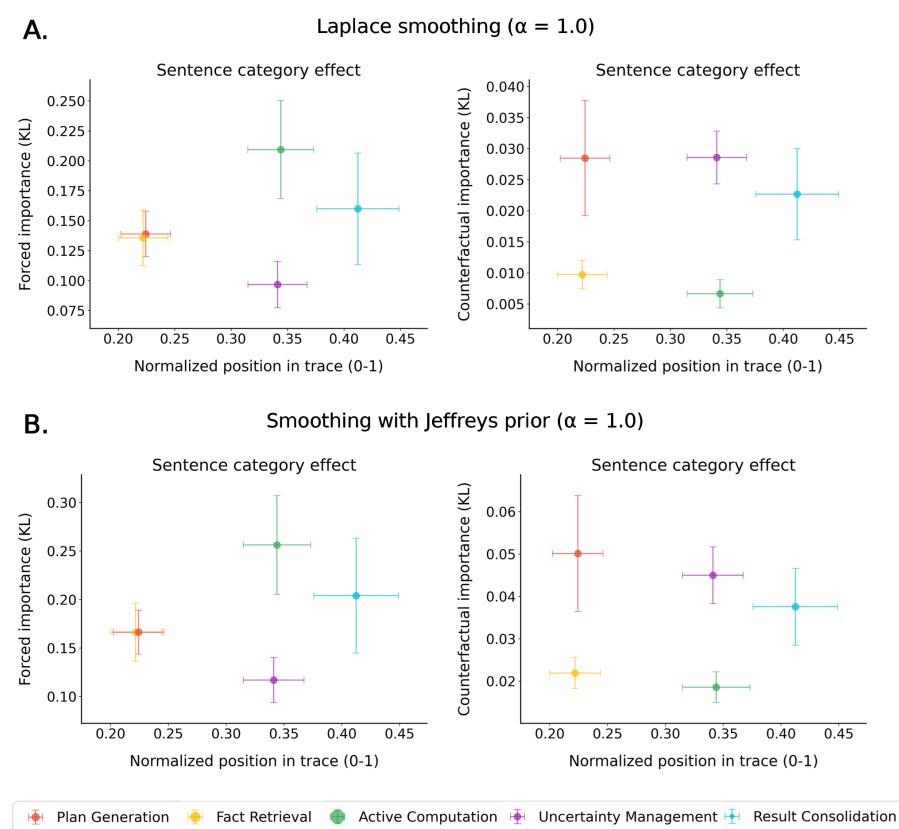

Figure 15: This is a variant of Figure 3, now performed with smoothing. Smoothing was performed using either **(A)** $\alpha = 1.0$, Laplace smoothing, or **(B)** $\alpha = 0.5$, Jeffreys prior.

## H  COUNTERFACTUAL VERSUS RESAMPLING IMPORTANCE

The resampling importance metric introduced in Section 2.3 treats all resampled sentences as equally informative, but different sentence types may exhibit varying degrees of **overdetermination** during resampling. Overdetermination occurs when resampled sentences $T_i$ are frequently similar to the original sentence $S_i$ (i.e., $T_i \approx S_i$), indicating that the reasoning context strongly constrains what can be expressed at that position. We present empirical evidence that counterfactual importance is a more nuanced measure by accounting for semantic divergence in resampled content.

Some sentences are more overdetermined than others. Figure 16A shows that *uncertainty management* and *plan generation* sentences produce semantically different alternatives in a large proportion of resamples, while *active computation* and *problem setup* sentences show lower divergence rates.

The transition matrix in Figure 16B shows how sentence categories change under resampling. For instance, *uncertainty management* and *active computation* sentences are usually replaced by sentences of the same category, whereas *plan generation* and *fact retrieval* sentences are more often resampled into a variety of other categories.

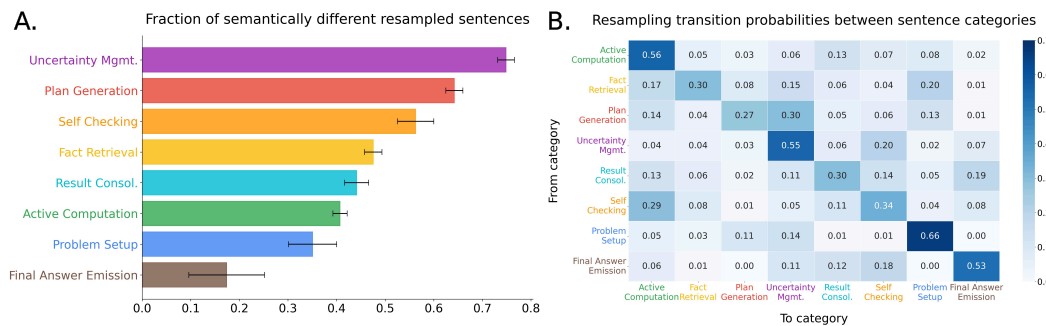

Figure 16: (**A**) Fraction of semantically different resampled sentences by category, showing that *uncertainty management* and *plan generation* sentences produce more divergent alternatives when resampled. (**B**) Transition probabilities between original and resampled sentence categories.

These resampling behaviors create systematic differences between our counterfactual and resampling importance metrics. Figure 17 demonstrates that the relationship between the two metrics varies substantially across sentences and sentence categories. The counterfactual importance metric aims to address overdetermination by explicitly filtering for semantically different resamples, providing a more targeted measure of causal influence. In contrast, the resampling metric potentially overestimates the importance of sentences that consistently produce similar content when resampled.

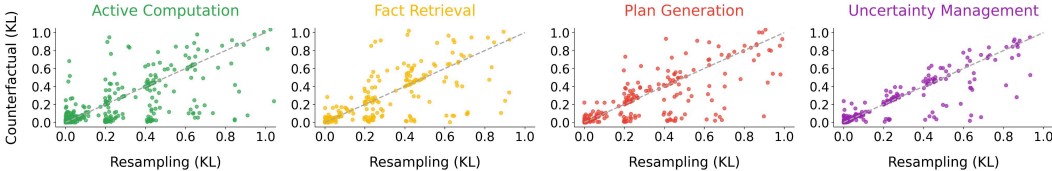

Figure 17: Comparison between counterfactual and resampling importance metrics across sentence categories. Each point represents a single sentence and the dashed gray line is the $y = x$ line.

However, the counterfactual importance metric can yield high-variance estimates when the number of semantically divergent resampled sentences is low (e.g., $< 10$), as the conditional probability estimates become less reliable with limited data. Alongside the limitations discussed in Section 8, this represents another constraint of our approach that future work should investigate further.

## I   SIMULATING INTERPRETATIONS FROM READING COT TEXT

To address whether our techniques for quantifying sentence importance merely recover insights that are obvious from reading the CoT text, we conducted an experiment simulating a human evaluator. We employed an LLM judge (Claude-4.5-Sonnet) to read the 40 MATH reasoning traces and predict the causal importance of specific sentences. We used the scores for two analyses: First, we measured the correlation between the judge's predicted importance and our empirical metrics (Counterfactual and Forced-Answer Importance). Second, we aggregated the judge's ratings against our sentence taxonomy—mirroring the analysis in the main text—to determine which sentence types (e.g., *Plan Generation* vs. *Active Computation*) a reader intuitively perceives as important.

### I.1   METHODOLOGY

For every sentence in our MATH dataset ($N = 6474$), we provided the LLM judge with the full reasoning trace up to that point. We tested two distinct prompting strategies (prompts in Table 3):

1. **Naive Causal Relevance:** We asked the judge to rate (0-10), "how much do you expect that the marked sentence is causally relevant to arriving at the correct final answer?"

2. **Mechanism-Aware Relevance:** We explicitly prompted it to consider the mechanics of autoregressive generation. We asked the model to consider "restarting the reasoning from that point" and to "think about how a given sentence influences the likelihood of future sentences appearing" before assigning a 0-10 rating.

Table 3: System prompts used for the LLM Judge evaluation. Both prompts included the full problem and the solution with the target sentence highlighted.

| Prompt Strategy | Prompt Template |
| --- | --- |
| **Naive Causal Relevance** | You are evaluating the importance of individual sentences in a mathematical reasoning trace. Below is a chain-of-thought solution to a math problem. One sentence is marked with **[THIS SENTENCE]** tags.

Problem: {problem}
Solution: {marked_solution}

Question: On a scale of 0 to 10, how much do you expect that the marked sentence is causally relevant to arriving at the correct final answer?
- 0 means completely irrelevant (removing it would have no impact)
- 10 means critically essential (removing it would definitely prevent getting the correct answer)

Provide ONLY a single number from 0 to 10 as your response. |
| **Mechanism-Aware Relevance** | You are evaluating the importance of individual sentences in a mathematical reasoning trace. Below is a chain-of-thought solution to a math problem. One sentence is marked with **[THIS SENTENCE]** tags.

Problem: {problem}
Solution: {marked_solution}

Question: On a scale of 0 to 10, how much do you expect that the marked sentence is causally relevant to arriving at the correct final answer?
- 0 means completely irrelevant (removing and restarting the reasoning from that point would have no impact)
- 10 means critically essential (removing and restarting the reasoning from that point would definitely prevent getting the correct answer)

Think about how a given sentence influences the likelihood of future sentences appearing. Provide ONLY a single number from 0 to 10 as your response. |

## I.2 CORRELATION RESULTS

We computed the Pearson correlation between the LLM judge's ratings and our empirically derived importance metrics. We found that the judge's importance scores did not meaningfully correlate with the ground-truth causal impact measured by our resampling method.

- **Correlation with Counterfactual Importance:** The judge's ratings showed near-zero correlation with our counterfactual importance metric for both the naive prompt ($r = -.03$) and the mechanism-aware prompt ($r = -.04$). This suggests that the sentences that actually drive the model's downstream trajectory are not intuitively obvious even to a strong frontier model analyzing the text.

- **Correlation with Forced-Answer Importance:** Interestingly, the judge's ratings showed a weak positive correlation with the forced-answer importance baseline for both prompting conditions ($r = .14 - .16$).

## I.3 DIFFERENCES ACROSS SENTENCE CATEGORIES

As shown in Figure 18, the divergence stems from what the judge values. The LLM judge tends to assign high importance to `active_computation`. In contrast, our resampling method identifies `plan_generation` and `uncertainty_management` steps as having the largest causal impact. Although there exists no single "ground truth" that all of these methods are attempting to predict, these comparisons demonstrate that our counterfactual technique identifies new insights and covers different mechanisms by considering how a sentence influences the downstream CoT.

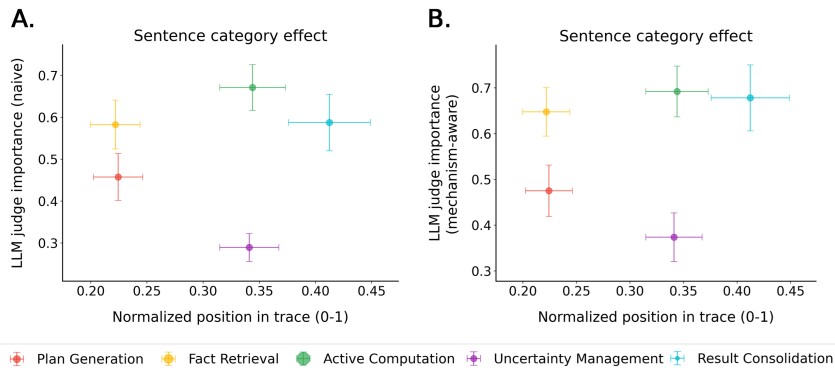

Figure 18: Importance scores assigned by the LLM judge, averaged for each sentence category.

## J ADDITIONAL RESAMPLING RESULTS

Figure 19 presents mean counterfactual importances across all eight taxonomic categories for R1-Distill-Qwen-14B, extending the main text results (Figure 3) which showed only the five most frequent sentence types. The expanded view includes three additional categories with lower frequencies. *Problem setup* sentences occur predominantly at trace beginnings (mean normalized position $\approx 0.1$) with moderate-high counterfactual importance. *Self checking* sentences tend to occur in the second-half of the traces and show lower counterfactual importance. *Final answer emission* sentences appear late in traces (mean normalized position $\approx 0.9$) and show the lowest counterfactual importance. The patterns observed in the five-category analysis remain consistent when examining the full taxonomy.

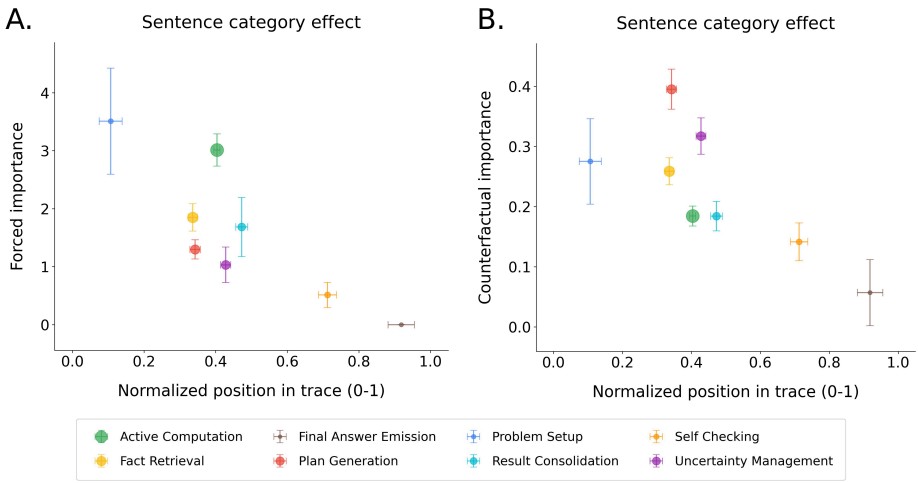

Figure 19: The mean of each sentence category for (**A**) forced-answer importance and (**B**) counterfactual importance for R1-Distill-Qwen-14B, per the resampling method, plotted against the sentence category's mean position in the reasoning trace. All sentence types are shown.

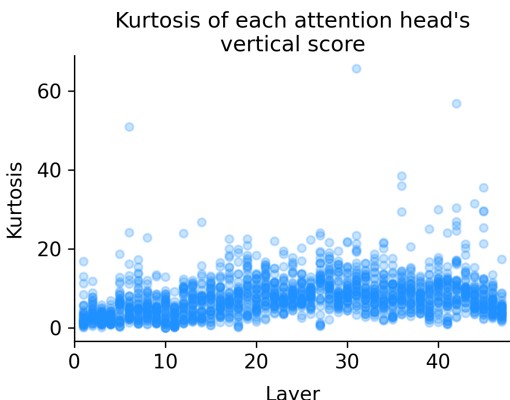

Figure 20: This scatterplot shows the kurtosis of each head's vertical-attention score, organized by layer. There is an upward trend across layers and a strong uptick among some late-layer heads.

## K  ADDITIONAL RECEIVER HEAD INFORMATION

Receiver heads – heads receiving high kurtosis scores – are more common in late layers (Figure 20). Examples of receiver heads are shown in Figure 21, showing how the highest kurtosis head consistently narrows attention on particular sentences, and Figure 22, showing how there exist many heads that narrow attention on particular sentences.

## L  REASONING VERSUS BASE MODEL DIFFERENCES IN RECEIVER HEADS

Attentional narrowing toward particular sentences may be a feature specifically of reasoning models. We submitted the reasoning traces to a base model version of Qwen-14B and identified receiver heads. For both models, we sorted all sentences by their mean receiver-head score using the 16 attention heads with the highest kurtoses. The highest percentile sentences received greater attention by the reasoning model - e.g., the highest-percentile sentences receive 1.8x more attention via top-16 heads in the reasoning model compared to the base model (Figure 23). Additionally, lower percentile sentences receive less attention through the top-16 heads. This conclusion is somewhat tenuous, as no base-model difference is seen when this result is tested using R1-Distill-Llama-8B. Nonetheless, based on the Qwen-14B data, it appears the model has learned to narrow its attention toward particular sentences.

## M  EFFECTS OF ABLATING RECEIVER HEADS

To test the causal hypothesis that the receiver heads identified in Section 4 are functionally important for reasoning, we performed an experiment ablating receiver heads and evaluating how this impact's model accuracy. This intervention is designed to measure the direct impact of removing these heads on task performance and to evaluate the possibility that they may be more important than typical heads.

### M.1  METHODOLOGY

We continue to use problems from the MATH dataset. We selected 32 problems where the non-ablated model achieves 10-90% accuracy on average. For each problem, we ran R1-Distil-Qwen-14B sixteen times, while allowing the model to output up to $2^{16}$ (16,384) tokens. Responses that did not produce an answer by that point were marked as incorrect.

We compared the effect of ablating 128 attention heads (approx. 7% of all heads), 256 heads (approx. 13%), or 512 heads (approx. 27%). The ablation strategies were:

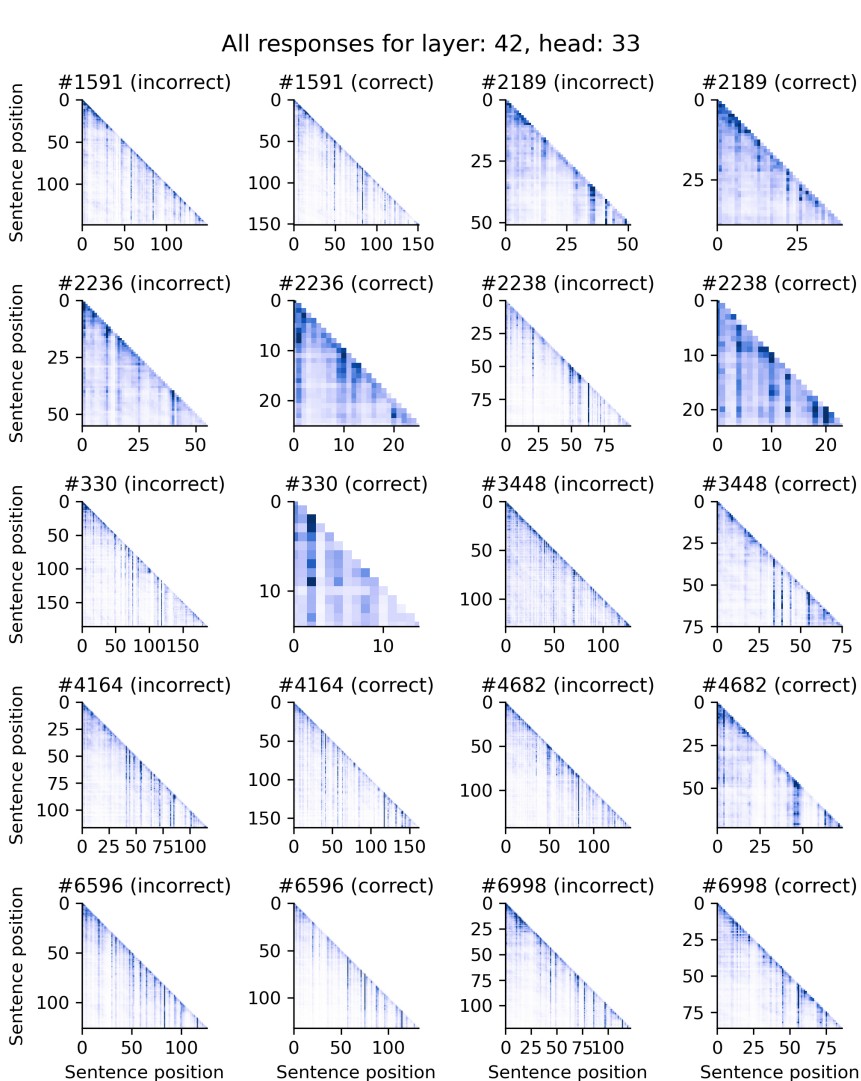

Figure 21: The attention weight matrices for the receiver head with the highest kurtosis score are shown here for twenty of the forty responses (selected arbitrarily based on the first twenty processed). The coloring was defined such that the darkest navy corresponds to values surpassing 99.5th percentile value of each matrix. White is zero.

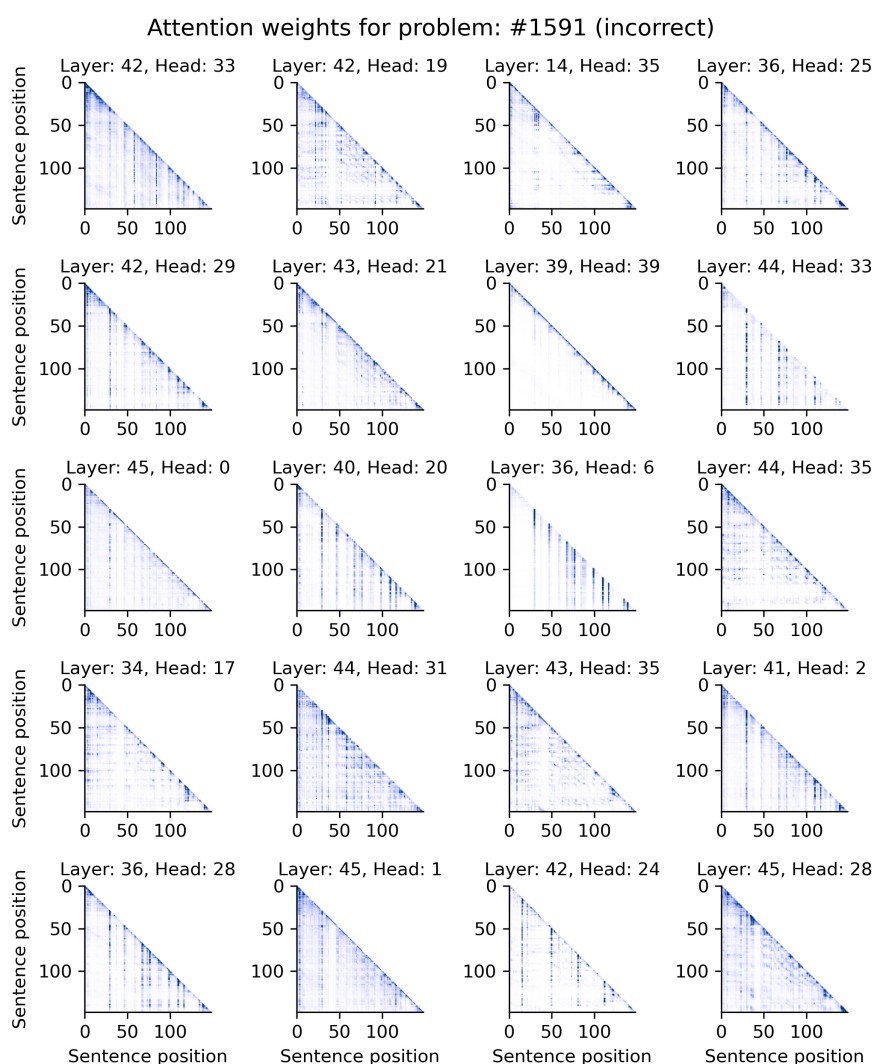

Figure 22: The attention weight matrices for response #1591 (incorrect) are shown here for the 20 attention heads yielding the highest kurtosis score across all responses. No effort was taken to "cherry-pick" responses showing prominent receiver head patterns; we are showing #1591 (incorrect) because it corresponded to the alphabetically earliest problem number among the ten problems analyzed (correct/incorrect chosen randomly). The coloring was defined such that the darkest navy corresponds to values surpassing 99.5th percentile value of each matrix. White is zero.

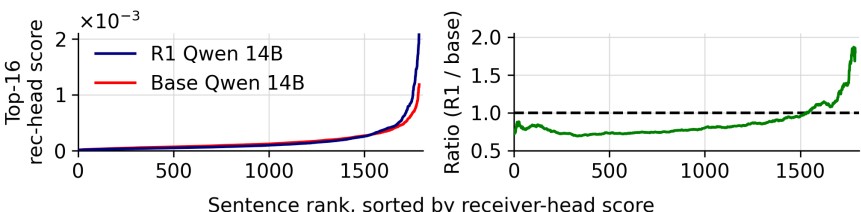

Figure 23: The navy and red lines on the left show the receiver-head scores assigned to sentences, averaged across the 16 heads with the highest kurtoses. The green lines on the right represent the ratio of the navy and blue lines for a given sentence rank. Sentences with high receiver head scores receive more attention in the reasoning model compared to the base model.

1. **Receiver head ablation:** We ablated the top-N heads with the highest average kurtosis scores.
2. **Random non-receiver (control) ablation:** For each layer where $k$ receiver heads were ablated, we ablated $k$ heads chosen randomly from the set of heads not selected from that same layer. This ensures a matched comparison with no overlap.

Note that receiver heads are more common in late layers (see above, Figure 20). By ensuring that both conditions included an equal number of heads from each layer (rather than selecting 128, 256, or 512 heads randomly across all layers), this ensures that differences cannot be explained simply by differences in the layers selected.

In the 512-head ablation condition, a majority of attention heads in some late layers were marked as receiver heads. For these layers, the non-receiver control condition was modified to ablate the corresponding number of heads with the lowest kurtosis scores to ensure a valid comparison set. For instance, if 60% of layer 43 heads are in the top-512, then the control condition included the 60% with the lowest kurtosis score, meaning that there is 20% overlap for that layer.

## M.2 RESULTS AND DISCUSSION

Our experiments show that a large number of heads must be ablated to induce a significant drop in performance compared to the baseline level of accuracy (baseline = 64.1%, 95% CI: [56.0%, 72.1%]). Regardless of whether receiver heads or non-receiver heads are targeted, ablating 128 heads produces differences in accuracy that insignificantly differ from baseline accuracy, and ablating 256 heads still produces only a small drop in accuracy (Table 4).[1]

Table 4: Answer accuracy on MATH problems for different self-attention-head ablation conditions. The brackets show the 95% confidence interval for each accuracy estimate.

| Heads Ablated | Receiver heads | Random heads |
|:---:|:---:|:---:|
| 256 | 48.8% [39.3%, 58.3%] | 52.7% [43.0%, 62.5%] |
| 512 | 27.7% [17.2%, 38.2%] | 37.3% [27.5%, 47.1%] |

The importance of receiver heads emerges when a large number of heads are ablated. When ablating 512 heads (over a quarter of the model's 1920 heads), targeting receiver heads caused performance to fall to 28% accuracy. Removing the same number of control heads resulted in a less severe drop to 37% accuracy. There is a significant difference between these percentages ($t[31] = 2.55, p = .02$), suggesting receiver heads are more critical for reasoning than other heads.

As mentioned, this analysis treats responses as incorrect if they do not produce a final answer by 16,384 tokens. If the analysis is changed to instead simply omit those responses entirely from the analysis, there remains a significant difference in accuracy when ablating top-512 receiver heads (29% accuracy) versus random non-receiver heads (39% accuracy) ($t[31] = 2.66, p = .02$). Hence, regardless of whether non-completed responses are marked as incorrect or ignored, ablating receiver heads is found to exert a larger impact on model accuracy than ablating random non-receiver heads.

## N    KL CAUSAL GRAPH PSEUDOCODE

This pseudocode outlines the procedure for computing a sentence-to-sentence causal graph for a given chain-of-thought (CoT). The algorithm works by systematically masking each source sentence and measuring the resulting change in the model's predictions (logits) for all subsequent target sentences. The sentence–sentence impact is quantified as the average log-KL divergence across a target sentence's tokens, which is then normalized against the average impact from all prior sentences.

---

[1]We are not aware of prior studies on attention head ablation for models generating long chain-of-thought reasoning, making it difficult to establish what is a typical number of heads to ablate. Potentially, a large number is necessary because the long reasoning traces (sometimes exceeding 10,000 tokens) provide extensive opportunities for error correction and compensatory computation.

This last normalization step effectively accounts for differences in target sentences' average entropy, which may vary widely and can hamper studying differences between target sentences.

Masking can be performed either by suppressing attention toward the source sentence or omitting the sentence entirely; the former preserves positional embedding information, while the latter may be computationally cheaper and easier to implement (e.g., with serverless providers). If masking is done by omitting sentence $i$ from the CoT, rather than suppressing attention toward sentence $i$, this will impact sentence $j$'s token positions across the CoT and masked CoT, which should be accounted for.

---

**Algorithm 1** GetCausalMatrix(CoT, Model)

---

1: Initialize CAUSAL_MATRIX $\in \mathbb{R}^{M \times M} \leftarrow 0$       $\triangleright$ $M$ = number of sentences in CoT
2: LOGITS_BASE $\leftarrow$ FORWARD_PASS(CoT, Model)       $\triangleright$ shape: (tokens, vocabulary)
3: **for** $i = 0$ **to** $M - 1$ **do**       $\triangleright$ source sentence
4:      CoT$_{\text{masked}} \leftarrow$ MASK_SOURCE(CoT, $i$)
5:      LOGITS_MASKED $\leftarrow$ FORWARD_PASS(CoT$_{\text{masked}}$, Model)
6:      **for** $j = i + 1$ **to** $M - 1$ **do**       $\triangleright$ target sentence
7:          TOKENS_J $\leftarrow$ SENTENCE_TOKENS(CoT$_{\text{masked}}$, $j$)
8:          TOTAL_KL $\leftarrow 0$
9:          **for each** $k \in$ TOKENS_J **do**
10:             KL $\leftarrow$ KLDIVERGENCE(LOGITS_BASE[$k$], LOGITS_MASKED[$k$])
11:             TOTAL_KL $\leftarrow$ TOTAL_KL + $\log$(KL)
12:          **end for**
13:          CAUSAL_MATRIX[$i, j$] $\leftarrow$ TOTAL_KL/|TOKENS_J|
14:      **end for**
15: **end for**
16:                     $\triangleright$ Normalize each target column by the mean over prior sources
17: **for** $j = 0$ **to** $M - 1$ **do**
18:      $\mu \leftarrow$ MEAN(CAUSAL_MATRIX[$0\!:\!j, j$])
19:      CAUSAL_MATRIX[$0\!:\!j, j$] $\leftarrow$ CAUSAL_MATRIX[$0\!:\!j, j$] $- \mu$
20: **end for**
21: **return** CAUSAL_MATRIX

---

**Algorithm 2** KLDivergence(LOGITS_P, LOGITS_Q)

---

1: $\log p \leftarrow$ LOGITS_P $-$ LOG_SUM_EXP(LOGITS_P)       $\triangleright$ log-softmax
2: $\log q \leftarrow$ LOGITS_Q $-$ LOG_SUM_EXP(LOGITS_Q)
3: $p \leftarrow \exp(\log p)$
4: KL $\leftarrow 0$
5: **for each** vocabulary index $v$ **do**
6:      KL $\leftarrow$ KL $+ p[v] \cdot \big( \log p[v] - \log q[v] \big)$
7: **end for**
8: **return** KL

---

## O   SENTENCE-TO-SENTENCE COUNTERFACTUAL IMPORTANCE

We extend our counterfactual resampling framework (section 3.2) to quantify each sentence's influence on each future sentence. Further below, we describe how this measure's values for sentence-sentence links correlate with the values generated via our section 5 method, masking sentences and measuring the impact on later sentences' logits.

### O.1   COUNTERFACTUAL SENTENCE-SENTENCE LINKAGE METHODS

We estimate the counterfactual importance of sentence $S_i$ on a future sentence $S^{\text{Fut.}}$ formally with:

$$\text{importance}(S_i \rightarrow S^{\text{Fut.}}) = \mathbb{P}(S^{\text{Fut.}} \in_{\approx} \{S_i, \ldots, S_M\}) - \mathbb{P}(S^{\text{Fut.}} \in_{\approx} \{T_i, \ldots, T_N\} | T_i \not\approx S_i) \quad (1)$$

Intuitively, on the right-hand side of Equation (1), the first term is the probability that a future sentence $S^{\text{Fut.}}$ will semantically occur given that $S_i$ was present in the trace, and the second term is the corresponding probability when $S_i$ is resampled with a non-equivalent sentence. A positive score indicates that sentence $S_i$ increases the likelihood of producing $S^{\text{Fut.}}$ (i.e., $S_i$ upregulates $S^{\text{Fut.}}$), while a negative score suggests that it suppresses or inhibits it. To be clear, this technique relies on full autoregressive rollouts rather than teacher-forced probabilities.

In this context $S^{\text{Fut.}}$ semantically occurs if, when we extract the sentences and identify the best candidate match for $S^{\text{Fut.}}$ using cosine similarity between sentence embeddings, it has greater than $0.8$ cosine similarity (i.e., the median value in our dataset) to that sentence. Pseudocode for estimating sentence-to-sentence importance and empirical values of this metric can be found in Section H.

Beyond measuring individual sentence importance, our framework quantifies causal dependencies between specific sentence pairs within reasoning traces. Figure 24 displays the sentence-to-sentence importance matrix for problem #2236 (incorrect) (*"Each page number of a 488-page book is printed one time in the book. The first page is page 1 and the last page is page 488. When printing all of the page numbers, how many more 4's are printed than 8's?"*), showing how individual sentences influence downstream reasoning steps. Below we list a few illustrative cases.

- **12-PG → 16-PG.** The planning in sentence 12 (*"1. Count the number of 4's in the units place across all page numbers"*) raises the probability that the model produces sentence 16 (*"Starting with the 4's."*) by $0.39$. A plan statement triggers a subordinate planning step.

- **8-FR, 9-PG, 12-PG, 14-PG → 32-UM.** The uncertainty management in sentence 32 (*"However, I need to check if 440-449 is fully included."*) receives sizeable positive influence from several earlier sentences: 8-FR ($+0.11$), 9-PG ($+0.06$), 12-PG ($+0.12$), 14-PG ($+0.10$). This forms the dense horizontal band at row index 32.

- **39-RC $\not\to$ 83-UM.** The result consolidation in sentence 32 (*"Now, summing up all the 4's: - Units: 48 - Tens: 50 - Hundreds: 89. Total 4's = 48 + 50 + 89 = 187."*) decreases the likelihood (i.e., inhibits) of 83-UM (*"Wait, but just to be thorough, let me check the hundreds place for 4's again."*) by $0.22$.

- **52-AC $\not\to$ 65-SC.** The computation in sentence 52 (*"The first four blocks 80-89, 180-189, 280-289, 380-389 each contribute 10 eights in the tens place."*) decreases the likelihood of 65-SC (*"Let me go through each step again to make sure I didn't make a mistake."*) by $0.16$.

- **63-AC → 64-UM, 65-SC, 69-SC, 75-SC, 83-UM, 86-SC.** The computation in sentence 63 (*"So, the difference is 187 – 98 = 89."*) propagates forward, increasing the likelihood of 64-UM ($+0.24$), 65-SC ($+0.17$), 69-SC ($+0.16$), 75-SC ($+0.28$), 83-UM ($0.23$), and 86-SC ($0.16$). This forms the dense vertical band originating from column index 63.

- **64-UM → 65-SC, 69-SC, 75-SC, 83-UM, 86-SC.** The uncertainty management in sentence 64 (*"Wait, that seems quite a large difference."*) further amplifies the same downstream block: 65-SC ($+0.32$), 69-SC ($+0.25$), 75-SC ($+0.26$), 83-UM ($0.25$), and 86-SC ($0.25$).

- **83-UM → 86-SC, 90-FAE.** Even very late checks matter. Sentence 83 (*"Wait, but just to be thorough, let me check the hundreds place for 4's again."*) increases the chance of 86-SC (*"Correct. And for the tens place..."*) by $0.43$ and of the final answer in 90-FAE by $0.41$.

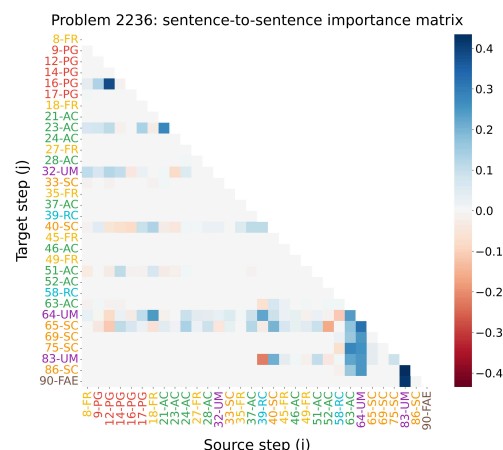

Figure 24: Sentence-to-sentence importance matrix for the 32 most important sentences in problem #2236 (incorrect), selected based on total outgoing and incoming importance. Each cell $(i, j)$ shows the causal importance of sentence $i$ on sentence $j$, calculated as the difference in the probability sentence $j$ semantically occurs ($> 0.8$ cosine similarity) when sentence $i$ is present versus resampled.

We provide the following pseudocode for estimating sentence-to-sentence importance:

```
Input: Sentence index i, target sentence index j (where j > i),
       threshold t = 0.8
Output: Importance score importance(i -> j)

1. Get rollouts R_keep where sentence i was kept (resampling from i+1)
2. Get rollouts R_remove where sentence i was removed (resampling from i)

3. For each rollout r in R_keep:
   a. Extract all sentences S_r from rollout r
   b. Find best matching sentence to target sentence j:
      - Compute sentence embeddings
      - Calculate cosine similarity between each s in S_r and target j
      - Select sentence with highest similarity if similarity >= t
   c. Add to matches_keep if valid match found

4. For each rollout r in R_remove:
   a. Extract all sentences S_r from rollout r
   b. Find best matching sentence to target sentence j
     (same process as step 3b)
   c. Add to matches_remove if valid match found

5. Calculate match rates:
   match_rate_keep = |matches_keep| / |R_keep|
   match_rate_remove = |matches_remove| / |R_remove|

6. Return importance(i -> j) = match_rate_keep - match_rate_remove
```

### O.2 CORRELATIONS WITH THE RESAMPLING-BASED IMPORTANCE MATRIX

The sentence-masking matrix values correlate with those of the resampling-method matrix. Specifically, the two matrices were positively correlated for 90% of reasoning traces (mean $r = .20$, 95% CI: [.12, .27]); a correlation was computed separately for each CoT and then averaged, and this average significantly surpasses zero per a one-sample t-test ($p < .001$). This association is stronger when considering only cases fewer than five sentences apart in the reasoning trace, which may better track

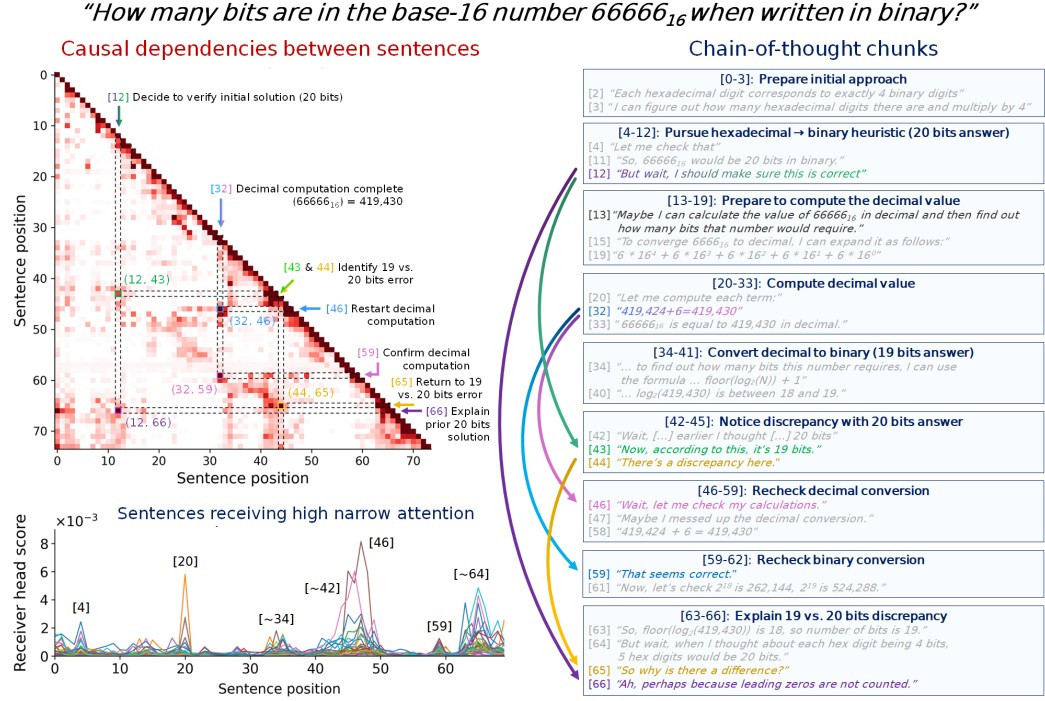

Figure 25: Case study: problem #4682 (correct). Red matrix shows the effect of suppressing one sentence (x-axis) on a future sentence (y-axis). Darker colors indicate higher values. Bottom-left line plot shows the average attention toward each sentence by all subsequent sentences via the top-32 receiver heads (32 attention heads with the highest kurtosis score). Flowchart summarizes the model's CoT with chunks defined around key sentences receiving high attention via receiver heads. Sentence 13 is emphasized as it has high counterfactual importance per the resampling method (see Figure 2A).

direct rather than indirect effects represented by the resampling method (mean $r = .34$ [.27, .40]). The magnitudes of these correlations are substantial, given that the two measures capture partially different aspects of causality and the resampling measure itself contains stochastic noise. Hence, these results give weight to the validity of the resampling approach, whose precision we leverage for the forthcoming case study.

## P    IN DEPTH CASE STUDY

The presented techniques cover different aspects of attribution within a reasoning trace. Building on the case-study conclusions from our resampling approach (section 2.4), we study the model's CoT here by focusing on receiver heads and sentence-sentence links (Figure 25) (see above, Section A, for the full transcript).

### P.1    RECEIVER HEADS

The trajectory toward the final correct answer can be understood as a series of computational chunks (see flowchart in Figure 25). First, the model prepares a formula for converting $66666_{16}$ to decimal (sentences 13-19). Next, the model computes the answer to that formula, finding that $66666_{16}$ is 419,430 in decimal (sentences 20-33). The model subsequently converts that number to binary by putting forth another formula and solving it, $floor(log_2(419,430)) + 1 = 19$, to derive that the answer is "19 bits" (sentences 34-41). The model then notes a discrepancy with the earlier 20-bit solution (sentences 42-45). The model hence initiates new computations that verify that it computed the decimal value of $66666_{16}$ correctly (sentences 46-58) and that it computed the binary conversion accurately (sentences 59-62). Equipped with this increased certainty about 19-bit answer, the model

discovers why its initial 20-bit idea was incorrect: *"because leading zeros are not counted"* (Sentence 66). This overall narrative is based on our analysis of attention patterns (section 4): Receiver attention heads pinpoint sentences initiating computations or stating key conclusions, thereby segmenting the reasoning trace into seemingly meaningful chunks (Figure 25).

## P.2 ATTENTION SUPPRESSION

Along with being organized into computational chunks, the reasoning displays a scaffold related to sentence-sentence dependencies (Figure 25). One piece of this structure is a self-correction pattern involving an incorrect proposal, a detected discrepancy, and a final resolution. Specifically, the model initially proposes an incorrect answer of "20 bits", which it decides to recheck (sentence 12). This leads to a discrepancy with the "19 bits" answer computed via decimal conversion (sentences 43 & 44). After rechecking its arithmetic supporting the "19 bit" answer, the model returns to the discrepancy (sentence 65) and then produces an explanation for why the "20 bits" answer is incorrect (sentence 66). This can be seen as a tentative CoT circuit, where two conclusions conflict to produce a discrepancy, which in turn encourages the model to resolve the discrepancy. Within this wide-spanning scaffold, there exist further dependencies, corresponding to verifying an earlier computation. Specifically, the model finishes computing the decimal value of $66666_{16}$ as 419,430 (sentence 32), later decides to verify that decimal conversion (sentence 46), and finally confirms that the original value is correct (sentence 59). This can be seen as further indication of CoT circuitry.

We identified these linkages based on the attention-suppression matrix (section 5), which contains local maxima at these linkages ($12 \rightarrow 43$, $43 \rightarrow 65$, $12 \rightarrow 66$; $32 \rightarrow 46$, $32 \rightarrow 59$). Notice that many of the sentences pinpointed by the attention-suppression technique overlap with the sentences receiving high attention from receiver heads. Adding to the receiver-head conclusions, the attention suppression technique shows how information flows between these key sentences that structure the reasoning trace.

## Q SENTENCE POSITION EFFECTS ON RECEIVER-HEAD SCORES

A sentence's position within the reasoning trace will tend to influence its measured receiver score.

As a reasoning trace progresses, the number of possible broadcasted sentences will necessarily increase. For instance, by sentence 20, there might be only two broadcasted sentences (each receiving 50% of attention from sentences 21-29), whereas by sentence 100, there could be ten broadcasted sentences (each receiving 10% of attention from sentences 101-109). As the sum of an attention weight row will sum to 1 (at the token level), later sentences will distribute their attention across a larger number of past sentences. This dilution of attention creates downward pressure on the receiver-head scores of later sentences. This is the case even though a receiver head score extends through all subsequent low-competition or high-competition periods. For example, broadcasting sentence 20 will face limited competition from receiving sentence 21-29 attention and high competition for sentences 101-109, whereas broadcasting sentence 100 will exclusively face high competition, pushing its score downward as broadcasting-sentence position increases.

There also exists a proximity effect on receiver-head scores that operates in the opposite direction of the above effect. Although broadcasted sentences are attended by all subsequent sentences to some degree, this will be more so the case for more recently subsequent sentences (e.g., receiving more attention from a sentence 10 sentences downstream than one 20 sentences downstream). For sentences late in the reasoning trace, the average distance to future sentences will be shorter. For example, if a reasoning trace contains 120 sentences, then sentence 100 will be at most 19 sentences apart from any given future sentence, whereas sentence 20 will be at most 99 sentences apart. To a degree, the analyses in the report account for proximity effects by ignoring the 4 sentences immediately proximal to a given sentence when calculating vertical-attention scores. However, this will not fully address proximity effects.

We see no reason why the downward pressure of sentence position on receiver-head scores (attention dilution) will be equal in magnitude to the upward pressure of sentence position (proximity effects).

For the preparation of the present report, we conducted exploratory analyses evaluating whether the above confounding factors invalidate any presented finding, and we did not find evidence that

this is the case. Thus, rather than pursuing some technique to account for the above pressures (e.g., linearly weighing attention weight matrices based on their position), we opted to only account for these factors in a minimal fashion by ignoring the attention among sentences just 4 sentences apart.

