# OpenReview forum: "Thought Anchors: Which LLM Reasoning Steps Matter?"
_ICLR.cc/2026/Conference — Submitted to ICLR 2026_

### Official Review · Reviewer_vUoT · 2025-10-28

**Soundness:** 2
**Presentation:** 1
**Contribution:** 2
**Rating:** 2
**Confidence:** 4

**Summary:**

This paper introduces two methods for interpreting chain-of-thought (CoT) reasoning in LLMs. The first method measures sentence importance by either forcing an LLM to generate an answer right after an intermediate sentence, or resampling multiple traces from a perturbed intermediate sentence. Quantitatively, the sentence importance is computed as the KL divergence between the original and resampled answer distributions. The second method aims to interpret the interactions between pairs of sentences. The authors first demonstrate that certain attention heads narrow attention toward specific sentences, and then measure the causal links between sentences by taking the KL divergence between the token logits with versus without suppressing a sentence. It is observed that such causal links is dependent of the distance between sentences, with strong close-range links and weak long-range links. The authors also released their results as a website for fancy interactions with the interpretations.

**Strengths:**

1. Considering the prevalence of CoT reasoning, interpreting and understanding CoTs has a large impact in the community of LLMs.
2. The authors released their results as a fancy interactive website, showing that their methods can potentially reveal insights behind LLM reasoning. However, this website is based on results pre-computed by the authors. It’s better if users can submit their own samples and obtain results online or offline.

**Weaknesses:**

1. This paper doesn’t meet the standard of a scientific paper. Most of the paper is about the methods, but there is scarcely any systematic evaluation of the methods. For example, as stated in Line 119, the authors only used 10 questions to evaluate their methods, and even the full evaluation results can only be found on their website. This is definitely not sufficient for a paper in terms of the number of datasets or samples. Results in Figure 2-7 are more like qualitative or weakly quantitative results showing example output from the proposed methods, which are not enough to justify the proposed methods. I would suggest either comparing the proposed methods against baselines with quantitative experiments, or showing a bunch of lessons that we can observe only with the proposed method. Or maybe the authors should submit this paper to the demo track of some other conferences.
2. This paper is poorly written. There are too many design details, but not enough high-level structure connecting the details, which makes the paper hard to understand. The authors should clearly describe their motivations for each method they introduce, and explicitly state the contribution of each method. The title especially blurs the contribution of this paper. In fact, this paper doesn’t provide a thorough answer to the question in the title, but offers a tool to help answer that question. I would suggest changing it to “Visualize important steps and their interactions in LLM reasoning”. In the introduction, the second paragraph discussing why you realize your methods at sentence level seems to be very weird. Usually, we expect this paragraph to serve the goals and motivations of your method. I also don’t understand why you need to introduce forced answer importance given you showed that counterfactual importance is better in Figure 3. Also several section titles are either confusing or not well supported by evidence in the paper. For Sec 3, it’s not clear what the authors refer to by consistent patterns, since they are never discussed in that section. For Sec 4, the authors didn’t discuss why attention head is the mechanistic roots of importance. For Sec 6, what are the systematic differences you found apart from the connection between distance and causal links?

**Questions:**

1. Line 88: What’s the faithfulness of CoT text? How do you sidestep it?
2. Line 140, 143: $S$ → $S_i$
3. Line 143: Are $S_{i+1}$, …, $S_M$ resampled? If so, you’d better use another notation to distinguish them against the original sentences.
4. Line 153-161: This paragraph looks verbose for the main paper. Better shorten it and move the full version to appendix.
5. Line 188: describes → defines
6. Line 207: $p(A'_{S_i})$ is not defined
7. Line 208: How $\epsilon$ is used in the equation is not introduced.
8. Line 234-235: Section H didn’t directly shows that counterfactual importance is more useful than resampling importance. In my opinion, resampling importance measures how far the current prediction is from the average prediction, which also makes sense in some cases.
9. Line 293-295: On how many samples did you compute the correlation? If it’s only 10 problems, that may not be enough.
10. Line 309: $r=0.22$ is not very significant. Also “may minimally impact” in Line 312 seems to conflict with “exert a larger effect”.
11. Line 338-339: In the original counterfactual resampling, you resample $S_j$ to be $T_j$, right? Here you only resample $S_i$ to be $T_i$ and then evaluate the likelihood of following sentences $S_j$ in a teacher forcing way, right?
12. Line 398: What do you mean by “sharply structure the CoT” here?
13. Line 409: $M_{sentences}$ is not defined.
14. Line 426: What are the potential structural roles and their relationship to successful reasoning? Please be explicit.

---

> ### Author Response · Authors · 2025-11-23
>
> Thank you for your feedback, efforts, and kind words on the strengths of our paper. The present comment and the beginning of the next comment respond to each point mentioned in the first noted weakness.
>
> **Re: Reliability and doubling the sample size.** The primary claims we draw from our analyses of 10 questions (20 responses) is that sentences vary in their importance and that planning and uncertainty management are consistently important. However, our paper also draws claims based on analyses of 2,492 problems’ responses, where we show that sentence-sentence linkages are meaningful and also display systematic patterns, now associated with differences in problem domains and difficulty. Although the latter analyses cover more problems, we do not believe our initial results are any less reliable because the underlying effects are large and robust.
>
> To demonstrate these findings’ robustness, we collected data on 10 new questions (20 new responses), and each result reported in the text or shown in the figures remains statistically significant. Further, in our original report, we had replicated our findings using a different model (Appendi B), and as we describe in our response to the third comment by Reviewer c8CH, our findings on counterfactual importance remain significant when we vary key parameters of the analysis. This would all be extremely unlikely if our results were noise. We agree that studying just 10/20 questions is often unreasonable but is sufficient here. This is in large part because each response generates over a hundred sentences. Although sentences are not independent within-question, this provides a substantial volume of samples, bolsters effect sizes at the question level, and permits replicable findings.
>
> **Re: Quantitative nature of the results.** We respectfully disagree with this critique: Figures 2 and 6 are qualitative, but 3, 4, 5, and 7 are not. The results in Figures 2 and 6 reflect our methods’ outputs for a single CoT to illustrate our methods’ outputs and show their capacity to generate narrow insights. An important contribution of our method is the ability to generate narrow insights about a particular CoT, so we believe that qualitative figures added to our paper here are valuable. However, Figure 7 quantifies trends across 2,492 reasoning traces, finding robust differences linked to problem difficulty and subject domain, Figures 3 and 5 demonstrate systematic differences in metrics across sentence types aggregated over 40 CoTs, and Figure 4 displays aggregate attention patterns across the network (1,920 attention heads) for the full dataset (40 CoTs). The analyses overall provide the quantitative foundation that our methods are tapping into generalizable properties of CoTs.
>
> **Re: Clarification on the shown baseline.** In the reviewer’s second comment, they ask why we “introduce forced-answer importance.” We apologize for the confusion, but this was meant as a baseline comparison. Forced-answer importance — sometimes called “early answering” — is a widely used technique in prior papers attempting to interpret CoT (Lanham et al., 2023; Radhakrishan et al., 2023; Parcalabescu & Frank, 2024; Tanneru et al., 2024; Wang et al., 2025). We updated the paper to include more citations to these earlier studies to make it clear that forced-answer importance is an existing baseline technique. Our results demonstrate that this standard method focuses on the role of active-computation steps but totally ignores the importance of planning or uncertainty management steps, which can shift the downstream CoT trajectory.
>
> **Re: New baseline comparison.** Consistent with this reviewer’s suggestions and the first comment by Reviewer c8CH, we have added an additional point of comparison for our counterfactual importance metric. To simulate the takeaways about reasoning that a human would gather from reading a CoT, we used an LLM judge and asked it to try and predict each sentence’s importance based on reading the text. We attempted two different LLM-judge prompts, but the judges’ assigned importance scores still showed virtually no correlation with the true counterfactual importance we measured (r = -.03 and r = -.04). We also find that the judge considers active-computation statements to be most important, like the older forced-answer importance method finds. We added these results to Appendix I.
>
> There exists no single “ground truth” interpretability score that all of these methods attempt to predict, but rather they cover different mechanisms. We argue that the phenomena captured by our techniques are meaningful and produce new insights. A priori our counterfactual importance method should capture how some sentences can shape the downstream trajectory, and this claim is consistent with our findings linking our technique to plan-generation and uncertainty-management sentences. The alternative techniques do not cover this mechanism.

---

> ### Author Response · Authors · 2025-11-23
>
> **Re: lessons we can observe only with the proposed method.** Our case studies are meant to demonstrate the capacity of our techniques for this aim, but going beyond this is our aim for a different paper. We discuss this in our response to Reviewer c8CH (comment #1) and Reviewer Hw1d (comment #4). In a separate submission, we use these tools to make specific insights about open questions. We have separated this into two papers, as each alone is already quite pushing on space limitations. It seems most appropriate to keep the present report as a detailed overview of the methods and only on general conceptual claims about the structure of CoT. In this paper, we demonstrate the value of our methods with more general takeaways, such as demonstrating the large impact of high-level structuring statements (plan generation and uncertainty management) on a CoT and the model’s final answer. But we would be happy to provide a link to an anonymised copy of the follow-up work if helpful.
>
> The next batch of responses engages with to each point made in the second noted weakness.
>
> **Re: Writing for high-level structure.** We have edited the manuscript to include a brief summary paragraph at the start of each section presenting results. These summaries explicitly state the motivation for the upcoming method and its specific contribution to the overall argument, helping guide the reader through the logical progression of the paper. We have also decreased somewhat the degree of minute details mentioned.
>
> **Re: Clarifying our goals in the second paragraph.** We updated the second paragraph of the introduction to instead focus on the goals and motivations of our method: It now explicitly states that our primary aim is to identify which steps in a CoT have outsized causal impact on the final answer and to develop a general toolkit for mapping their interactions. We also briefly preview how the three components of our toolkit (black-box resampling, receiver heads, and sentence-masking graphs) each serve this goal, so that the later methodological sections are clearly motivated from the outset.
>
> **Re: Forced-answer importance.** As clarified in our response above, we include forced-answer importance specifically to serve as a baseline comparison. By contrasting it with our resampling method, we demonstrate that this technique fails to identify how sentences can be causally important by influencing the downstream CoT. We have revised the text to make this comparative purpose clearer.
>
> **Re: Section 3 (“consistent patterns”).** By "consistent” or “systematic” patterns, we refer to the robust finding that specific taxonomic categories (Plan Generation, Uncertainty Management) consistently yield high importance scores across the dataset, whereas others (Active Computation) do not. This is a pattern that generalizes across problems. We updated the text to be clearer about what we mean.
>
> **Re: Section 4 ("mechanistic roots").** We argue that "receiver heads" provide concrete evidence for some sentences within a CoT being particularly important, and we provide initial signs of a plausible mechanism for how they may influence downstream sentences. On the validity of “receiver heads” at all, we found a strong split-half reliability in their kurtosis scores across problems (r = .84), indicating that these heads consistently narrow attention on specific sentences. Furthermore, different receiver heads consistently attend to the same target sentences (mean r = .56), which is consistent with sentence importance generally. While we acknowledge that this part of our work is preliminary, this section seems like a valuable addition that further advances our argument that CoT contains key impactful moments. We edited the section title to “Mechanistic evidence for sentence importance”, which seems more consistent with this argument.
>
> **Re: Section 6 ("systematic differences").** We argue that these findings explain why chain-of-thought provides performance uplift in some domains but not others and, more generally, how different domains vary in terms of the CoT they elicit. Hence, the types of patterns we find are indeed “systematic differences.” For example, the domains that benefit most from CoT, like mathematics, display particularly strong sequential dependencies, and this is a consistent pattern across CoTs in this area, identified by our sentence-masking technique. We edited the text in this section to be clearer.

---

> ### Author Response · Authors · 2025-11-23
>
> The reviewer pointed out some small typos or terms that needed to be defined. We appreciate these, and we have addressed them. Below, we focus on the more conceptual comments or ones requiring clarification.
>
> **Re: faithfulness of CoT text.** Our underlying methods effectively sidestep the faithfulness debate by providing an objective measure of causal influence that is independent of the textual content. However, our empirical results arguably provide strong evidence for CoT text being at least partly faithful. When we categorize sentences based purely on their textual function (e.g., "Plan Generation" vs. "Active Computation"), we find that these labels consistently predict high values in our objective measures (receiver-head scores and counterfactual importance). This alignment demonstrates that the text at least somewhat reflects the specific computational mechanisms guiding the model's output.
>
> **Re: sentence–sentence counterfactual method (no teacher forcing).** This method used to measure sentence-sentence links based on counterfactual importance does not use teacher forcing. Our procedure determines whether Sj​ appears anywhere in the downstream trajectory after replacing Si with counterfactual Ti. Specifically, the procedure is:
>
> 1. **Intervention.** We replace sentence Si​ with a counterfactual Ti​.
> 2. **Autoregressive generation.** We allow the model to generate the remainder of the chain-of-thought freely (full rollout), without forcing it to follow the original path.
> 3. **Semantic search.** We scan the newly generated rollout to see if the target sentence Sj​ (or a semantic equivalent, defined by cosine similarity > 0.8) re-emerges.
>
> This metric captures causal necessity: whether removing Si​ causes Sj​ to disappear from the future trace. We have revised Appendix N to explicitly state that this relies on full autoregressive rollouts rather than teacher-forced probabilities.
>
> **Re: meaning of "sharply structure the CoT."** By "sharply structure," we refer to strong causal links, and in this context we are describing how planning steps in mathematical reasoning may precisely determine what comes next with little possibility of variability. For instance, a model may plan, “I must compute the value of 6^5.” It continues “6 × 6 = 36,” “36 × 6 = 216,” etc. Each token becomes highly determined by the preceding planning step, and the tokens’ logprobs go up considerably if the planning step is masked. We edited our description to be clearer.
>
> **Re: structural roles of planning/uncertainty sentences in success.** When we refer to “structure”, we are describing how some sentences influence other sentences, and here we are specifically discussing how the distance of plan-generation and uncertainty-management sentences’ impact may vary as a function of question difficulty. Difficult problems may require integrating information across multiple parts that each needs to be reasoned through, and so plan-generation sentences may exert causal effects across wider distances. Difficult problems may also produce discrepancies in reasoning noted in uncertainty-management sentences; as the Section 6 case study shows, discrepancies are tied to causal links between an original statement and a later discrepancy with that statement. We updated this part of the text to state these hypothesized roles explicitly.
>
> **Re: length and role of the main case-study paragraph.** We agree that this paragraph is a bit long, but we have found that it is difficult to shorten without making the case study entirely unclear, as understanding the underlying math problem and the logic used to solve it requires being precise. Although the entire case study could be moved to the appendix, we believe this would weaken the paper. The case study plays an important role by illustrating concretely how our tools can be used to dissect a single problem in depth.
>
> **Re: definition and use of epsilon.** We added Appendix G which formalizes the use of epsilon to avoid division by zero when discussing Laplace smoothing, where epsilon is replaced with a value of 1.0 (and becomes now typically referred to as alpha).
>
> **Re: counterfactual vs resampling importance.** This is a fair point, and we agree with the reviewer that both metrics have valid, distinct use cases. We have clarified in the revision that the resampling method would be more useful in some cases.
>
> **Re: effective sample size for correlation estimates.** Sorry about this confusion! The correlation was measured across sentences for each CoT (e.g., if a CoT had 140 sentences, then this was a correlation among 140 points). Then, the correlation coefficients were averaged across the 40 CoTs, yielding a mean correlation of r = .56. The sample size is effectively very large.

---

> ### Author Response · Authors · 2025-11-23
>
> **Re: significance and interpretation of modest correlations.** The correlation of r = .22 was taken by averaging across multiple CoTs, and has a rather tight confidence interval, computed by taking the standard error across problems (95% CI = [.17, .30]); in 90% of problems, the sentence-masking matrix of sentence-sentence causal effects positively correlates (r > 0) with the matrix based on the counterfactual resampling method, and a one-sample t-test shows that this significantly surpasses zero (p < .001). Although this value is nominally small, this partly stems from a 'floor effect' in our data. Because most sentences in a reasoning trace are computational steps with little causal weight, the correlation metric is heavily penalized by noise within this large pool of low-importance sentences. About the discrepancy on line 312, thank you for catching, and we have corrected it.

---

### Official Review · Reviewer_Hw1d · 2025-10-31

**Soundness:** 3
**Presentation:** 3
**Contribution:** 3
**Rating:** 6
**Confidence:** 4

**Summary:**

The paper proposed methods to interpret the chain-of-thought of LLMs by: 1) repeatedly resampling from each sentence and measuring the counterfactual impact of the sentences; 2) analyzing the attention pattern of LLMs; 3) intervening the attention scores to see direct causal effects of a sentence on subsequent sentences.

**Strengths:**

The paper conducted a systematic analysis of the reasoning traces of LLMs by resampling and masking attention weights at a sentence-sentence level. Several interesting observations are made in the paper, e.g., correlation between problem difficulty and range of sentence-level links in the reasoning traces.

**Weaknesses:**

See questions.

**Questions:**

1. Is the analysis in Section 2 to 4 all based on one particular problem (the 66666 problem)? It would be clearer if somewhere in the paper clearly states what analysis is done on what set of problems.
2. What information is Figure 3 trying to deliver?
3. In Section E, do the training labels of the classifier come from prompting the LLM? The section is trying to show that the LLM annotated sentence type is accurate/meaningful by showing that using the annotation as training labels, we can easily learn a good sentence classifier? If this is the case, I think the evidence for saying the LLM annotation is accurate may not be sufficient, as it only show that LLM did good clustering job, and the clustering can be reconstructed using the information in the reasoning model's hidden representation, but how accurate is the sentence type labeling cannot be inferred from this experiment. Some more examples or some human verification may be more helpful.
4. Figure 3 seems to suggest that each type of the sentences forms a continuous piece of text, and there is no interleave between sentence types. This doesn't seem to be the case in the web demo, where sentence types are interleaved.
5. While the findings in the paper are interesting, I find it hard to link the findings to practical guidelines. Can you give some examples of what scenarios may benefit from the findings in this paper?

---

> ### Author Response · Authors · 2025-11-23
>
> Thank you for your feedback, efforts, and kind words on the strengths of our paper.
>
> **Re: which sections are case studies vs aggregate analyses.** Sorry about this confusion! Section 2 and Section 5 mostly focus on the 66666 problem, treating it as a case study using methods. However, all of the remaining sections look at aggregate trends across numerous problems. We updated the paper to be clearer about this and to include “case study” in sections that present case studies.
>
> **Re: interpretation and intent of Figure 3.** This figure demonstrates systematic differences in importance across sentence types (y-axis) while plotting their average position (x-axis) to demonstrate that these importance differences are not merely artifacts of when a sentence appears in the trace. For example, while "Plan Generation" and "Problem Setup" both tend to appear early, only the former has high counterfactual importance. We apologize for the confusion and have updated the figure caption to explicitly state that the x-axis represents a distributional average across the dataset and is shown to rule out concerns about positional confounds, rather than implying that sentence types occur in continuous, non-interleaved blocks.
>
> **Re: validating LLM-based sentence-type labels.** The classifier analysis in Appendix E is indeed intended to validate that the LLM-generated labels correspond to distinct, linearly separable mechanistic states within the reasoning model’s residual stream. The high F1 score (0.71) confirms that categories like "Plan Generation" correspond to real differences in the model's internal computation, rather than just surface-level text features. However, as the reviewer states, this result does not validate the specific labels themselves.
>
> We have verified the labels ourselves as part of developing the LLM labeler prompt, and our released dataset provides the labels assigned to each sentence. We appreciate the reviewer’s suggestion to provide more examples, and we have added Table 2 to Appendix C, which provides ten randomly selected examples for each category to demonstrate the qualitative accuracy of the taxonomy.
>
> **Re: practical scenarios and use cases for our toolkit.** As we describe near the end of our response to comment #1 from Reviewer c8CH, this toolkit is meant to help with unpacking LLM reasoning in key scenarios and identifying general tendencies in a model’s reasoning. The tools and principles identified are poised to be useful for distinguishing between stated motives and causal drivers. The toolkit is not meant for active monitoring of all LLM outputs.
>
> We focus on specific scenarios with open questions in a parallel paper also under review for ICLR 2026. For instance, we apply our tools to studying LLM blackmailing ("agentic misalignment") scenarios, which have received much attention. We found that while models often output alarming text like "I must ensure my own survival," our resampling method revealed these sentences had negligible causal impact—identifying them as post-hoc rationalizations rather than true drivers. Similarly, in resume screening, we identified implicit biases (e.g., penalizing "overqualification") that were not explicitly stated. These applications demonstrate the toolkit's utility for debugging "silent" logic failures and auditing model reasoning. We have separated this into two papers, as each alone is already quite pushing on space limitations and we believe the case studies here provide initial demonstrations, enough for this manuscript to stand on its own merits. But we would be happy to provide a link to an anonymised version of the follow-up paper if useful.
>
> We updated the Discussion to briefly describe our vision for these methods’ usefulness.

---

### Official Review · Reviewer_d1j2 · 2025-11-01

**Soundness:** 4
**Presentation:** 3
**Contribution:** 3
**Rating:** 6
**Confidence:** 2

**Summary:**

This paper introduces a framework for identifying which sentences in LLM’s CoT reasoning are most causally important for the final answer. Using counterfactual resampling, attention-head analysis, and causal masking, it shows that planning and uncertainty-management sentences tend to steer the reasoning trajectory. These high-impact sentences attract focused attention from specific “receiver heads” in later layers, forming structural anchors for reasoning.

**Strengths:**

1. The paper offers a fine-grained interpretability analysis by examining reasoning at the sentence level rather than the token level, allowing a clearer view of how individual reasoning steps contribute to the overall thought process.


2. It introduces a sentence-masking method to study causal dependencies between reasoning steps, providing a thorough and systematic analysis of how earlier sentences influence later ones in the reasoning process.


3. This work releases an open-source interactive visualization tool that displays reasoning traces, sentence importance, and causal links between sentences. This tool makes it easy to observe how specific reasoning steps influence others.

**Weaknesses:**

1. The analysis assumes clean sentence segmentation and treats each sentence as an independent reasoning unit, which may not hold in more complicated reasoning contexts where boundaries are fuzzy.

2. The "thought anchor" concept is derived from a very limited dataset: just 20 reasoning traces from 10 math problems. Furthermore, the study only selected problems the model can solve 25-75% of the time, so the findings may not generalize to problems the model consistently fails or solves easily.

**Questions:**

Could you elaborate on how this framework might handle 'thought anchors' that don't fit neatly into a single sentence boundary? For example, how would it account for a critical reasoning step that might be a single key phrase within a much longer sentence, or, conversely, a larger concept that is built up across several sentences?

---

> ### Author Response · Authors · 2025-11-23
>
> Thank you for your feedback, efforts, and kind words on the strengths of our paper.
>
> **Re: sentence-level units vs fuzzy reasoning boundaries.** We view sentence segmentation as a practical heuristic and a robust starting point for decomposing reasoning. Empirically, initial results (Figure 2) suggest that sentences strike an optimal balance, avoiding the noise and cost of token-level analysis while offering significantly higher precision than paragraphs. Beyond just practicality, there are also studies showing that the punctuation marks ending sentences often serve key functional roles, compressing a large degree of information and acting as boundaries (Razzhigaev et al., 2025; Chauhan et al., 2025). Hence, we view the focus on sentences as an effective first hypothesis for the breakdown of a CoT.
>
> Regarding the reviewer's specific scenarios: for concepts spanning multiple sentences, we suspect that most of the causal importance will generally be captured by the initiation sentence that sets the trajectory. Conversely, if a critical insight is contained within a single sub-phrase, our techniques alone would be insufficient for identifying this. Nonetheless, our methods would still flag the full sentence as important and set the stage for targeted post-hoc inspections on portions of the statement.
>
> We updated the Introduction to provide more justification for our focus on sentences while also mentioning these points about sub-sentence or multi-sentence reasoning steps.
>
> **Re: dataset size, difficulty range, and generalization.** We appreciate this feedback. We see our analyses of sentence importance as analogous to other mechanistic interpretability efforts that intensively study a fairly small number of prompts, such as work on cross-layer transcoders (Ameisen et al., 2025). This can produce reliable results, and our findings are indeed robust. To show this, we have doubled the dataset for the resampling/receiver-head experiments to 40 reasoning traces across 20 problems. We performed the original analyses again and replicated our original results, identifying plan generation and uncertainty management sentences as being highly counterfactually important and highly attended to by receiver-head (p < .05 in all comparisons to active computation and fact retrieval sentences). Hence, these findings on sentence importance using 40 reasoning traces seem no less reliable than our other findings on sentence-sentence causal links covering 2,492 reasoning traces.
>
> Regarding the reviewer’s second point, our study focused on difficult problems because (i) for the resampling analysis, variation in the final answer distribution is the basis of measuring importance, and (ii) for the sentence-sentence causal link analysis, where we attempted to predict a question’s accuracy based on its graph properties, using problems with 25–75% accuracy imparts the most statistical power. We see no reason why our methods would not generalize to studying odd behaviors (e.g., <5% or >95% one way), although we agree that some reasoning motifs may be different in those contexts. For instance, a problem could be difficult because it has many points for subtle computational errors that would be difficult to later correct; in this case, the sentences of a correct CoT would all have low counterfactual importance, but an incorrect CoT would just have one large spike in counterfactual importance where the error occurred.
>
> We updated the Discussion to briefly note this issue with conceptual generalization while retaining that the methods should still be useful.

---

### Official Review · Reviewer_c8CH · 2025-11-03

**Soundness:** 3
**Presentation:** 2
**Contribution:** 3
**Rating:** 4
**Confidence:** 3

**Summary:**

The paper proposes a sentence-level interpretability framework. It introduces thought anchors to expose CoT traces by computing resampling importance, and further characterizes relationships among anchors using attention-based analysis.

**Strengths:**

1. The paper proposes a sentence-level toolkit for analyzing CoT traces using both black-box and white-box methods.

2. Step-by-step case studies accompany each stage of the pipeline, improving clarity and reproducibility.

3. It introduces thought anchors—sentences identified by estimating per-sentence importance via resampling scoring—and further examines relationships among anchors through attention analysis.

**Weaknesses:**

1. Motivation clarity: The paper argues for interpreting CoT, but does not clearly explain why CoT text alone is insufficient, nor how the method advances beyond prior interpretability work.

2. Computational cost: Analyzing one CoT trace is expensive (e.g., ~100 resamples per sentence plus an auxiliary labeling model). Practicality at scale is unclear.

3. Ablations are limited: No systematic study of design choices (e.g., sentence attention: mean/last-token/concat; Counterfactual importance: similarity function selection and threshold).

4. Organization: Lacks an upfront system diagram and end-to-end workflow before diving into components.

5. Experimental consistency: Different datasets/models appear across sections (e.g. MATH dataset and Qwen-14B in Section 2.1 while MMLU and  Qwen3-30b-a3b in section 6.1), making comparisons hard.

6. Reproducibility: As a tooling paper, code and usage docs are essential but not clearly provided.

**Questions:**

1. In what concrete scenarios does CoT text fail to be self-explanatory, and how do thought anchors remedy those gaps?

2. How sensitive are the identified anchors to prompt formatting and to the number of resampling rollouts used?

3. How does the approach perform on large models tackling harder tasks that yield long CoT traces?

---

> ### Author Response · Authors · 2025-11-23
>
> Thank you for your feedback, efforts, and kind words on the strengths of our paper.
>
> **Re: why CoT text alone is insufficient.** We appreciate this opportunity to elaborate on our motivation. Prior research on “faithfulness” has shown that there is sometimes a disconnect between CoT text and the factors actually influencing a model’s final decision/answer (Turpin et al., 2023; Chen et al., 2025). Additionally, even when a CoT largely reflects why a model arrived at some solution, it can still be difficult to predict which reasoning steps were most causally relevant based on just reading the CoT text. Our case study demonstrates this, showing how the most causally relevant sentence is not the model discovering a problem in its initial calculations, but rather is the subsequent sentence where the model makes a new plan for how to check its answer. Overall, we believe that reading CoT text can be useful for generating a high-level hypothesis of how a model arrives at an answer but is limited for pinpointing the impact of specific computations -- the typical goal of interpretability research.
>
> We conducted an empirical study that simulates a human attempting to intuit importance from just reading a reasoning trace. For each sentence, we asked an LLM judge (Claude-4.5-Sonnet) to rate “on a scale of 0-to-10, how much do you expect that the marked sentence is causally relevant to arriving at the correct final answer?” Tested on all 6474 sentences from our MATH dataset, we find that the LLM ratings do not meaningfully correlate with counterfactual importance (r = -.03). We also tried prompting the model to explicitly consider “restarting the reasoning from that point” and to “think about how a given sentence influences the likelihood of future sentences appearing,” but the model’s predictions were still uncorrelated with counterfactual importance (r = -.04). The LLM judge’s predictions instead correlate with forced-answer importance (r = .14–.16), which does not account for how a sentence impacts the downstream reasoning trace. In general, the LLM judge focuses on active-computation steps and undersells uncertainty-management steps, which we show in the added figure (Figure 9). We added a more detailed description of this experiment and its results as Appendix I.
>
> Even beyond just reading a CoT text, prior interpretability methods are limited for studying CoTs. Our paper provides a comparison to arguably the most widely used existing method (forced-answer importance). We show how this existing technique fails to capture how some sentences influence the final answer by impacting the downstream CoT, but this is captured by our counterfactual importance metric.
>
> As an additional point of evidence regarding the utility of our techniques, in a separate paper that we have currently under review at ICLR, we leverage the tools developed here to draw insights about a specific domain where the present approach produces insights that run contrary to impressions from reading CoT text -- namely, we study LLM blackmailing (“agentic misalignment”) and show how statements about a model wanting to preserve itself actually have virtually no causal impact on its final decision, raising questions about the typical interpretation of this widely studied scenario. This separate paper is specifically on applications and generating theoretical findings about the scenarios themselves, rather than presenting methods and making general claims about the structure of CoT, as done for the present report. Naturally, due to anonymity constraints, we can’t link to this work, but we believe that our paper stands on its own merits regardless. We’d be happy to provide an anonymised copy of the subsequent paper if useful.
>
> **Re: computational cost and intended use cases.** We view this framework as a specialized set of tools for auditing specific chains of thought in key scenarios and/or for generating broad insights about a given model’s general reasoning tendencies. It is not feasible as a technique for real-time monitoring. In interpretability research or AI safety work more generally, it is common to perform intensive experiments to understand the factors at play in a specific scenario — e.g., building a dataset to fit a probe or conducting deep case studies on phenomena like agentic misalignment. Consequently, the cost itself is manageable for a researcher; the intensive CoT resampling analysis of 40 problems in our revised manuscript incurred a total API cost of roughly \$250, given \$0.20 per million output tokens. The analysis of sentence-sentence causal links across 2,492 problems was around \$100. This remains financially feasible for many researchers even when scaling up to the most intelligent open-source models (e.g., gpt-oss-120b or qwen-235b-a22b would be only roughly 2–3× more expensive).
>
> We updated the Discussion to describe the costs more precisely and detail our vision for this method’s use cases.

---

> ### Author Response · Authors · 2025-11-23
>
> **Re: ablations and robustness to design choices.** Thank you for bringing this up. We have expanded our analysis to demonstrate that our findings are robust to design choices and not artifacts of specific hyperparameters:
>
> 1. We re-evaluated our counterfactual importance results while varying cosine similarity thresholds (0.5, 0.8, 0.9) for defining semantic equivalence. Additionally, we tested different smoothing parameters for the KL divergence calculation (Laplace smoothing or smoothing with Jeffreys prior). Across these configurations, our core finding — *plan generation* and *uncertainty management* sentences are the primary causal anchors — remained statistically significant (p < .05). We added the results for different similarity thresholds as Appendix F and the results for different levels of smoothing as Appendix G.
>
> 2. For the counterfactual importance analysis, the reviewer also asks about our choice of similarity function, namely the use of a sentence-embeddings model and measurement of cosine similarity. The former choice seems fine to accept as a default, given its prevalence in the NLP literature when a fast representation of semantics is necessary. The use of cosine similarity as opposed to some other measure has an extremely small effect. For a given pair of embeddings across the whole dataset, the pair’s cosine similarity will correlate extremely strongly with its Pearson correlation, Spearman correlation, or Euclidean distance (|r| > .98).
>
> 3. We further verified that our results regarding attentional patterns remain significant and consistent regardless of the threshold used for identifying receiver heads. The strong attentional focus on planning and uncertainty management sentences holds whether we analyze the top-16, top-32, or top-64 heads with the highest kurtosis scores (ps < .001 in comparisons to active-computation or fact retrieval sentences). We updated the caption of Figure 5 to note this. We did not reanalyze the receiver head data further, given that it was a smaller portion of the report than the resampling analysis.
>
> **Re: paper organization and overall structure.** We appreciate the suggestion regarding organization. However, we wish to clarify that our work presents a modular toolkit of complementary methods—black-box resampling, white-box receiver heads, and causal masking—rather than a single sequential processing pipeline. These techniques are designed to be applied independently or in parallel to triangulate sentence importance from different angles (causal vs. mechanistic), rather than functioning as a linear workflow. We believe Figure 1 currently illustrates these components as distinct approaches effectively.
>
> In general, to aid in clarifying the structure of the paper, we have updated each empirical section so that it begins with a paragraph describing its goals and how it contributes to the wider manuscript.
>
> **Re: model/dataset consistency across sections.** We utilized a dual-model strategy to demonstrate generalizability and accommodate technical constraints. The large-scale masking analysis (Section 6) required a specific serverless API endpoint returning token logits, which was available for Qwen-30b-a3b but not for the Qwen-14b model used in our local, compute-intensive resampling experiments. We updated the manuscript to make this clearer. If the reviewer deems it critical, we are willing to re-perform the initial resampling analysis (Section 2) using Qwen-30b-a3b to ensure strict experimental uniformity.

---

> ### Author Response · Authors · 2025-11-23
>
> **Re: code release, reproducibility, and tooling.** Thank you for pointing this out. To clarify, we do not see our report entirely as a tooling paper because we additionally make several theoretical claims about CoT and provide novel methods. Nonetheless, we wholeheartedly want to support users interested in implementing these methods. We have released the data and code needed to reproduce our results:
> <https://anonymous.4open.science/r/thought-anchors-2CB0>
>
> Additionally, we have developed two Python packages to help users apply these techniques to their own investigations:
>
> 1. A package that aids in querying an API by efficiently organizing API outputs and ensuring CoT prefilling is done correctly: <https://anonymous.4open.science/r/rollouts-6C4D/>
> 2. A package that helps with dividing a CoT into sentences and finding the token range corresponding to each sentence; these functions may seem trivial but are actually very prone to errors that can drastically influence the results: <https://anonymous.4open.science/r/Sentences-1148>
>
> Before ICLR 2026, we additionally plan to prepare example scripts showing how the two packages can be used to implement a Thought-Anchors analysis on any user CoT. As part of this, we will also provide some technical suggestions and details that we found critical for implementation but that go more into the weeds than appropriate for an ICLR paper — e.g., discussing how, when splitting CoT into sentences, it is important to avoid trailing spaces after periods for tokenization reasons. This information will be linked to the released packages, and the packages will be linked from the camera ready.
>
> **Re: sensitivity to prompt formatting and rollout count.** Regarding prompt formatting, we suspect that slight differences in a prompt will not have a large impact compared to simply changing the initial seed used for the model’s generation, which will cause the base CoT to differ. However, this issue seems beyond the present scope. In the separate report mentioned, we nonetheless demonstrate consistency in what sentences are most important across reasoning traces to the same question (e.g., in an experiment on how LLMs evaluate resumes, we find consistent patterns in what sentences have the highest positive/negative effects on the evaluation).
>
> Regarding the number of resampling rollouts used, this would impact the degree of noise, but the confidence intervals in this context can be calculated straightforwardly: for simplicity and in final-answer accuracy, a 95% confidence interval for 100 resamples corresponds to at worst ±10%. We updated the Discussion to mention this interval.
>
> **Re: scaling to larger models and longer CoTs.** To clarify, the CoTs that we tested are fairly long. In our resampling experiments using R1-Distill-Qwen-14B, the average response length was 144.2 sentences (approx. 4,208 tokens). Similarly, in our large-scale MMLU analysis (Section 6) with Qwen3-30B-a3b, the average length was 90.1 sentences. These lengths are a product of our filtering criteria, targeting difficult problems (25–75 accuracy), which tend to generate longer CoTs.
>
> Regarding larger models, we acknowledge that scaling or differences in training could introduce emergent reasoning phenomena — e.g., potentially larger models are better at error correction, so it is less likely that any one sentence could produce a sharp decline in accuracy. We see no principled reason why the core mechanisms we identified or our methods would not still apply to studying these larger models. In the separate CoT-resampling paper under review that we mentioned in response to the first comment, we actually included analyses of DeepSeek-R1 and Qwen3-235b-a22b in our studies. Although that paper focused on different domains than math problems, our counterfactual importance measure was able to find dissociations in causal importance across sentence types in those models as well. Nonetheless, we updated the present paper’s Discussion to note the possibility of emergent differences when scaling up to larger models or longer CoTs.

---

### Author Response · Authors · 2025-11-23

Our paper identifies and introduces methods for studying thought anchors—sentences in an LLM chain-of-thought (CoT) that have outsized causal influence on downstream reasoning and the final answer. We present findings that shed light on the causal factors within a CoT influencing a model’s final answer and we provide new black-box and white-box interpretability methods.

Reviewers found the topic at hand to be important and highlighted several strengths:

- **Reviewer c8CH:** Noted that our use of “step-by-step case studies [accompanying] each stage of the pipeline, improving clarity and reproducibility.”
- **Reviewer d1j2:** Emphasized that the work offers a “fine-grained interpretability analysis” that allows “a clearer view of how individual reasoning steps contribute to the overall thought process.”
- **Reviewer Hw1d:** Found that our “systematic analysis” yielded “several interesting observations.”
- **Reviewer vUoT:** Pointed out that “interpreting and understanding CoTs has a large impact in the community of LLMs.”

While noting these positive aspects, the reviewers also provided several pieces of helpful feedback. We have worked hard to fully address each of their comments. Our major changes to the manuscript include:

- **Improving the paper’s structure and clarity.** We rewrote the early introduction to clearly state our goal (identifying CoT steps with outsized causal impact and mapping their interactions) and how each method contributes. We also added brief summary paragraphs at the start of each empirical section to improve clarity regarding the overarching structure of our investigation.
- **Increasing the dataset size, including robustness checks, and offering a new baseline comparison.** We doubled the size of the dataset we used for our resample/receiver-head analyses and replicated each of our initial findings. We also performed robustness checks showing that our conclusions are stable across multiple design choices (e.g., similarity thresholds, smoothing parameters, receiver-head selection), and we added a new baseline comparison against an LLM judge that attempts to predict sentence importance from CoT text alone. These findings bolster the conceptual contributions of our report, going beyond just presenting new methods.
- **Enhancing reproducibility and providing more tooling.** To help other researchers interested in these techniques, we released two Python packages to help with implementation of these methods.
- **Expanding the Discussion and detailing use cases.** We added numerous new points to our Discussion and we have clarified how our proposed techniques are well-suited for intensive off-line analyses of model behavior.

Taken together, these revisions sharpen the paper’s motivation and structure, strengthen our conclusions’ robustness, and clarify how the proposed methods can be applied in practice.

 In the uploaded PDF, we have indicated changes made to address these comments in blue. Below, we provide responses to the reviewers’ specific comments.

---

### Meta-Review · Area_Chair_VBk4 · 2025-12-20

**Summary:**

This paper introduces an interpretability framework combining black-box resampling and attention analysis to identify "thought anchor" sentences, that has causal impact on the final answer. One problem of this paper is that the evaluation sample size (only 40 traces) too small to support broad scientific claims about LLM reasoning mechanisms. Furthermore, the proposed method has high computational cost, which raises the concerns regarding whether the framework is useful in practice. Reviewers also have concerns about paper writing and presentation/evaluation of the method. Therefore, this paper does not yet meet the standard for acceptance.

**Reviewer Concerns:**

Addressed:
1. The reviewer questioned whether the results were artifacts of specific hyperparameters. The authors showed that their core findings are stable across differnt hyperparameters.
2. Reviewers questioned why reading the CoT text wasn't sufficient. The authors added a new baseline experiment using an "LLM Judge" to predict sentence importance, showing it had little correlation with their causal metric.
3. Reviewers noted the lack of code and usage documentation. The authors addressed this by releasing two Python packages and the full dataset/code repository.

Outstanding
1. Insufficient sample size. Although the authors doubled their dataset from 10 to 20 problems (40 traces), reviewers explicitly stated that this scale is still too small to support broad scientific claims about "LLM reasoning mechanisms" in general.
2. High computational cost: The reviewer raised concerns about the practicality of running ~100 resamples per sentence, which is time consuming.

**Reviewer Scores:**

Reviewer c8CH may not change the score because the authors did not address the computational cost concern.

Reviewer d1j2 may not change the score because the authors did not address the dataset too small concern.

Reviewer Hw1d may not change the score because it is already positive (6).

Reviewer vUoT may not change the score because he/she has strong opinion on paper writing and method evaluation.

---

### Decision · Program_Chairs · 2026-01-26

Reject